# Large scale and integrated platform for digital mass culture of anchorage dependent cells

Kyoung Won Cho[1,2,4], Seok Joo Kim[1,2,4], Jaemin Kim[1,2,4], Seuk Young Song [2,4], Wang Hee Lee[1,2], Liu Wang [3], Min Soh[1,2], Nanshu Lu[3], Taeghwan Hyeon [1,2], Byung-Soo Kim[2]* & Dae-Hyeong Kim[1,2]*

Industrial applications of anchorage-dependent cells require large-scale cell culture with multifunctional monitoring of culture conditions and control of cell behaviour. Here, we introduce a large-scale, integrated, and smart cell-culture platform (LISCCP) that facilitates digital mass culture of anchorage-dependent cells. LISCCP is devised through large-scale integration of ultrathin sensors and stimulator arrays in multiple layers. LISCCP provides real-time, 3D, and multimodal monitoring and localized control of the cultured cells, which thereby allows minimizing operation labour and maximizing cell culture performance. Wireless integration of multiple LISCCPs across multiple incubators further amplifies the culture scale and enables digital monitoring and local control of numerous culture layers, making the large-scale culture more efficient. Thus, LISCCP can transform conventional labour-intensive and high-cost cell cultures into efficient digital mass cell cultures. This platform could be useful for industrial applications of cell cultures such as in vitro toxicity testing of drugs and cosmetics and clinical scale production of cells for cell therapy.

[1] Center for Nanoparticle Research, Institute for Basic Science (IBS), Seoul 08826, Republic of Korea. [2] School of Chemical and Biological Engineering, Seoul National University, Seoul 08826, Republic of Korea. [3] Center for Mechanics of Solids, Structures, and Materials, Department of Aerospace Engineering and Engineering Mechanics, Texas Materials Institute, University of Texas at Austin, Austin, TX, USA. [4]These authors contributed equally: Kyoung Won Cho, Seok Joo Kim, Jaemin Kim, Seuk Young Song. *email: byungskim@snu.ac.kr; dkim98@snu.ac.kr

Industrial utilization of anchorage-dependent cells has been rapidly increasing especially for biomedical, pharmaceutical, and cosmetic applications. For the industrial utilization such as cell therapies and in vitro toxicity testing, a large number of anchorage-dependent cells need to be cultured. Generally, stacks of multiple culture plates in multiple incubators are used for large-scale culture of anchorage-dependent cells[1]. However, current large-scale cell cultures tend to be inefficiently monitored and controlled due to the lack of an automated system that can provide real-time and high-resolution monitoring and localized control of the culture.

Microscopy and impedance sensing[2] have been employed for monitoring cell cultures. To achieve microscopy-based monitoring of cellular activities such as proliferation and differentiation, cells are periodically taken outside of the incubators. However, this process increases risk of bacterial contamination and abrupt perturbations in culture conditions[3]. Also, sacrifice of cells is inevitable to observe cellular differentiation by immunostaining. Furthermore, such monitoring in large-scale cell cultures requires a considerable amount of trained manpower and is time-consuming. For more efficient monitoring of cell cultures, impedance sensors have been used. However, current systems equipped with impedance sensors still lack multifunctionality[4], wireless controllability[5], and multilayer array mapping capabilities[6,7] for large-scale cell culture. In addition, an integrated method to sense key culture conditions, such as pH, temperature, and potassium ion ($K^+$) concentrations, and to provide localized wireless electrical and thermal stimulations for control of cell proliferation and differentiation is not available yet.

Herein we present a large-scale, integrated, and smart cell culture platform (LISCCP) that facilitates the digital mass culture of anchorage-dependent cells. A series of multifunctional device integrations by employing multi-layer transfer of ultrathin device arrays and wireless connection of each platform (Supplementary Fig. 1a) lead to a dramatic increase in the number of integrated devices (Supplementary Fig. 1b). The proliferation and differentiation of cultured cells can be three-dimensional (3D) monitored in a digital manner and even promoted by local stimulations. The centralized culture monitoring and control enable dramatic reduction in the culture-operation labor, which can transform labor-intensive high-cost cell culture into efficient digital mass cell culture (e.g., Supplementary Fig. 2).

## Results
### Large-scale, integrated, and smart cell culture platform.
LISCCP (Fig. 1a) consists of four main components: (i) arrays of ultrathin sensors and stimulators, (ii) graphene oxide (GO)-coated and 3D-printed polylactic acid (PLA) substrates, (iii) wirelessly controllable data acquisition system composed of home-made printed circuit boards (PCBs) assembled on Arduino DUE boards, and (iv) an in situ culture medium exchange and circulation system. By using wirelessly integrated LISCCPs, the status and culture environment of the anchorage-dependent cells in 3D space[8] are continuously monitored, and appropriate stimulations are locally applied[9] to facilitate cellular proliferation and differentiation. Owing to wireless integration, such monitoring and control can be done at the remote position without removing the cells from the incubator.

The impedance, pH, $K^+$, and temperature sensors, as well as the thermal and electrical stimulators, are spatially distributed on the GO-coated stackable PLA substrate integrated with the wirelessly system (Fig. 1b left, see Supplementary Fig. 3 for detailed arrangement of individual devices). Arrays of ultrathin sensors and stimulators[10,11] are integrated on multiple layers encapsulated with epoxy (SU-8) and polyimide (PI; Fig. 1b right).

The total thickness of the device is ~11 μm. The thickness of each layer and the scanning electron microscopic (SEM) image are shown in Supplementary Fig. 4. The detailed fabrication procedures are described in Supplementary Fig. 5 and "Methods." Microscopic images of individual devices are shown in Fig. 1c–f.

The 3D-printed PLA substrate[12] is designed to have multiple perforations (1 mm × 1 mm) for easy penetration of the culture medium and for distribution of seeded cells throughout multiple layers, protruding hemispheres ($D = 1$ mm) to increase the surface area, and stackable pillars and holes at edges for multilayer assembly (Supplementary Fig. 6). The PLA substrates integrated with ultrathin sensors and stimulator arrays by transfer printing[13] are easily stacked into multilayer for mass culture of adherent cells (Fig. 1g shows a five-layer stack; inset shows the SEM image of protruding hemispheres).

The five-layer PLA substrate stack is housed in an acrylic case to organize the wireless systems and to standardize assembly dimensions. Multiple LISCCPs can be stacked for space efficiency (e.g., three units; Fig. 1h), which are placed inside the incubator for mass cell culture (Fig. 1i). Each LISCCP in the incubators is wirelessly integrated for large-scale monitoring and control of the cell culture (Fig. 1j). The process of the in situ medium exchange is shown in Supplementary Fig. 7.

### Characterization of substrates and devices.
Figure 2 describes details of the GO-coated stackable PLA substrate and characterization of sensors and stimulators. The engineered PLA substrates can be integrated into multilayer by assembling pillars on the top surface into holes at the bottom like Lego® blocks (Fig. 2a; inset shows magnified image of the joint). C2C12, a stable mouse myoblast cell line, is utilized as a model cell line to evaluate the cell-favoring environment of the engineered substrate and the GO coating. Compared to a single flat cell culture substrate (without protruding hemispheres and perforations), the five-layer integration of engineered substrates with protruding hemispheres and perforations leads to sixfold increase in the number of attached cells (Fig. 2b). The cell-attachable surface area of a single engineered substrate is higher than that of the flat substrate despite the loss of surface area from the perforations (Supplementary Fig. 8a). Because the perforations allowed easy nutrient and oxygen exchange[14], cells could be cultured in up to five stacks with ~100% cell viability even without medium circulation (Fig. 2c; see Supplementary Fig. 8b and Supplementary Note 1.1 for oxygen diffusion). The depth of the culture medium is critical to cell viability. The cell viability begins to decrease after the culture medium reached 10 mm in height and drops to ~50% when the height of the culture medium reaches 30 mm (Fig. 2d). See Supplementary Fig. 8c for the decreased viability of cells after stacking up to 10 layers. But this issue of decrease in cell viability by multi-layer integration over five layers can be solved by continuous medium circulation, which will be explained later.

The GO nanosheet is spray-coated onto the PLA substrate (Fig. 2e; inset shows the GO solution) to enhance cellular adhesion[15]. The atomic force microscopic (AFM) image of the GO-coated PLA substrate shows increased surface roughness (Fig. 2f), allowing cells to easily attach onto the substrate. The Raman spectrum of GO and the AFM image of the bare PLA substrate before GO coating are shown in Supplementary Fig. 9a, b, respectively. The SEM images of the GO-coated substrate before and after cell seeding (Fig. 2g) demonstrate cell adhesion on the substrates. The GO coating increased the cell adhesion by ~50% compared to glass or polystyrene substrates[16] (Fig. 2h). SEM images of cells at the curved edge of the perforation shows high cellular adhesion and proliferation stimulated by the GO coating (Supplementary Fig. 9c). The cell adhesion and

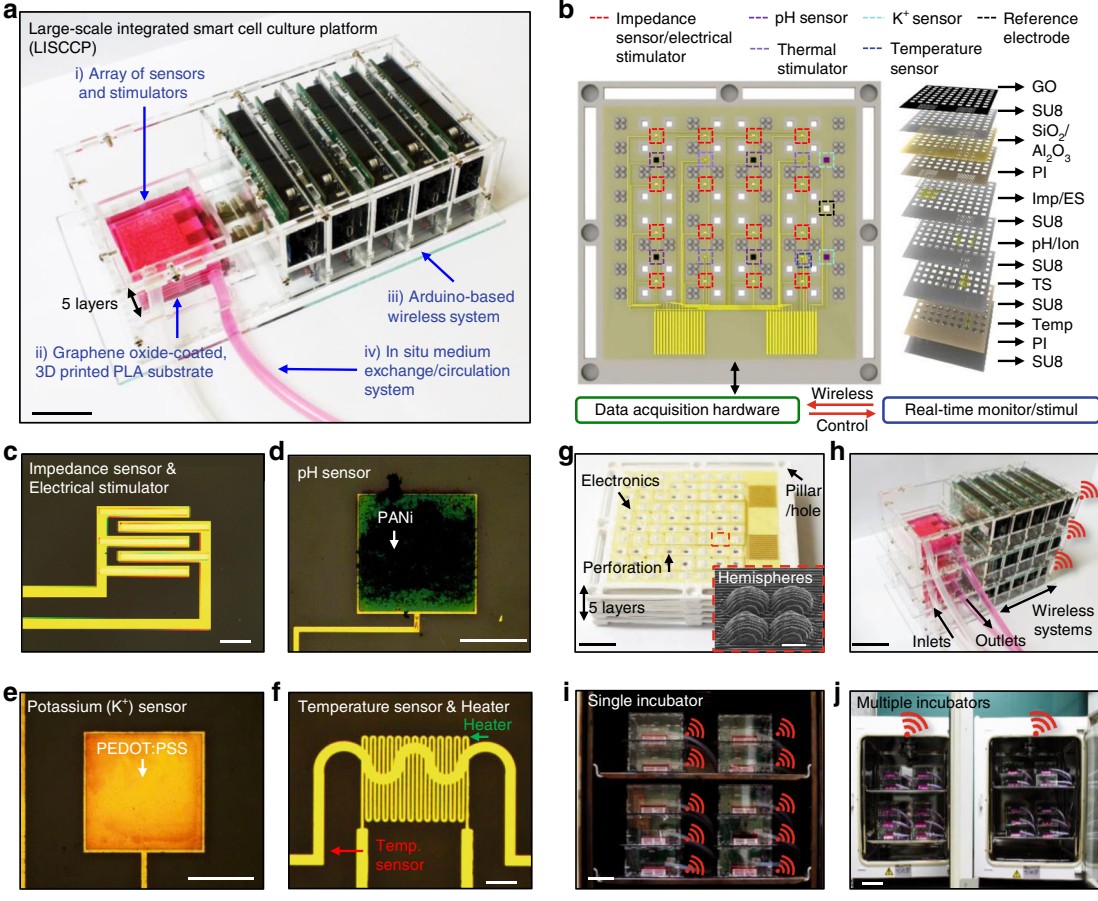

**Fig. 1** Large-scale, integrated, and smart cell culture platform. **a** Photograph showing LISCCP. Scale bar: 2 cm. **b** Schematic illustration of the ultrathin sensors and stimulators transfer-printed to the 3D engineered substrate to be integrated with the wireless control system (left) and the exploded view of the ultrathin device layers (right). **c–f** Optical microscopic image of the impedance sensor/electrical stimulator (**c**), pH sensor (**d**), K⁺ sensor (**e**), and temperature sensor and heater (**f**). Scale bars for **c**, **f**: 200 μm; for **d**, **e**: 500 μm. **g** Photograph showing the five-layer stacked 3D-printed engineered PLA substrates that were designed with protruding hemispheres (D = 1 mm, inset), perforations (1 mm × 1 mm), and stackable pillars and holes and integrated with the ultrathin electronics. Scale bar: 1 cm. Inset shows the SEM image of the protruded hemispheres. Scale bar: 500 μm. **h** Photograph of the 3D stack assembly of three LISCCPs. Scale bar: 5 cm. **i** Photograph of a $CO_2$ incubator filled with multi-stacked LISCCPs (a total of 10) for digital mass cell culture. Scale bar: 5 cm. **j** Photograph of two $CO_2$ incubators filled with multi-stacked LISCCPs wirelessly integrated with a single laptop. Scale bar: 10 cm

subsequent proliferation increase further as the concentration of GO used for PLA coating increases (Supplementary Fig. 9d). The cell proliferation increases as the GO concentration increases but saturates after the GO concentration reaches 200 mg 100 mL⁻¹ ethanol (Supplementary Fig. 9e). The GO can be patterned into line-and-space arrangements to induce cellular alignment[17] (Supplementary Fig. 9f) that can mimic in vivo environments of certain cells such as neural and muscle cells[18,19] to maintain their phenotypes.

Characterizations of sensors and stimulators are shown in Fig. 2i–l. The pH and K⁺ sensors measure open-circuit potential (OCP) at different concentrations of hydrogen ion (H⁺) in the pH range of 4–10 (Fig. 2i left) and K⁺ concentrations in the range of 1–32 mM (Fig. 2j left), respectively. Insets show calibration curves. These sensors can detect subtle pH changes between 6.95 and 7.95 (Fig. 2i right), and K⁺ changes between 0.125 and 2 mM (Fig. 2j right). The selectivity of pH and K⁺ sensors is confirmed with other cations such as $Mg^{2+}$, $Ca^{2+}$, and Na⁺ (Supplementary Fig. 10a, b).

Interdigitated electrodes serve as electrical stimulators and impedance sensors. The current–voltage relationship within the culture medium and the impedance at frequencies between 1 and 1,000,000 Hz are characterized in Fig. 2k. Insets show reproducibility. The resistance of the temperature sensor changes linearly

and sensitively with temperature variation (Fig. 2l left and its inset). The temperature of the heater increases linearly from 29 to 45 °C as the applied voltage increases from 2 to 8 V (Fig. 2l right). The long-term stability of all sensors is presented in Supplementary Fig. 10c–h.

**Wireless data acquisition and control system**. Figure 3 describes the data acquisition and control system for automated monitoring and localized stimulation and its algorithm. The arrays of sensors and stimulators are connected to a home-designed PCB, which is assembled on an Arduino DUE board composed of a 32-bit microcontroller (MCU) and a few digital–analog converters (DACs) (Fig. 3a). More detailed information is provided in Supplementary Fig. 11 (circuit diagram and PCB artwork), Supplementary Fig. 12 (installed components and functions), Supplementary Fig. 13 (LabVIEW block diagram), and Supplementary Fig. 14 (algorithm summary) with descriptions in Supplementary Note 1.2.

Figure 3b–e shows the flow chart illustrating the signal processing sequence for monitoring cell culture status and/or facilitating cell growth. For impedance measurement, the programmatically generated sinusoidal signals are delivered to the desired impedance sensor through multiplexing circuits

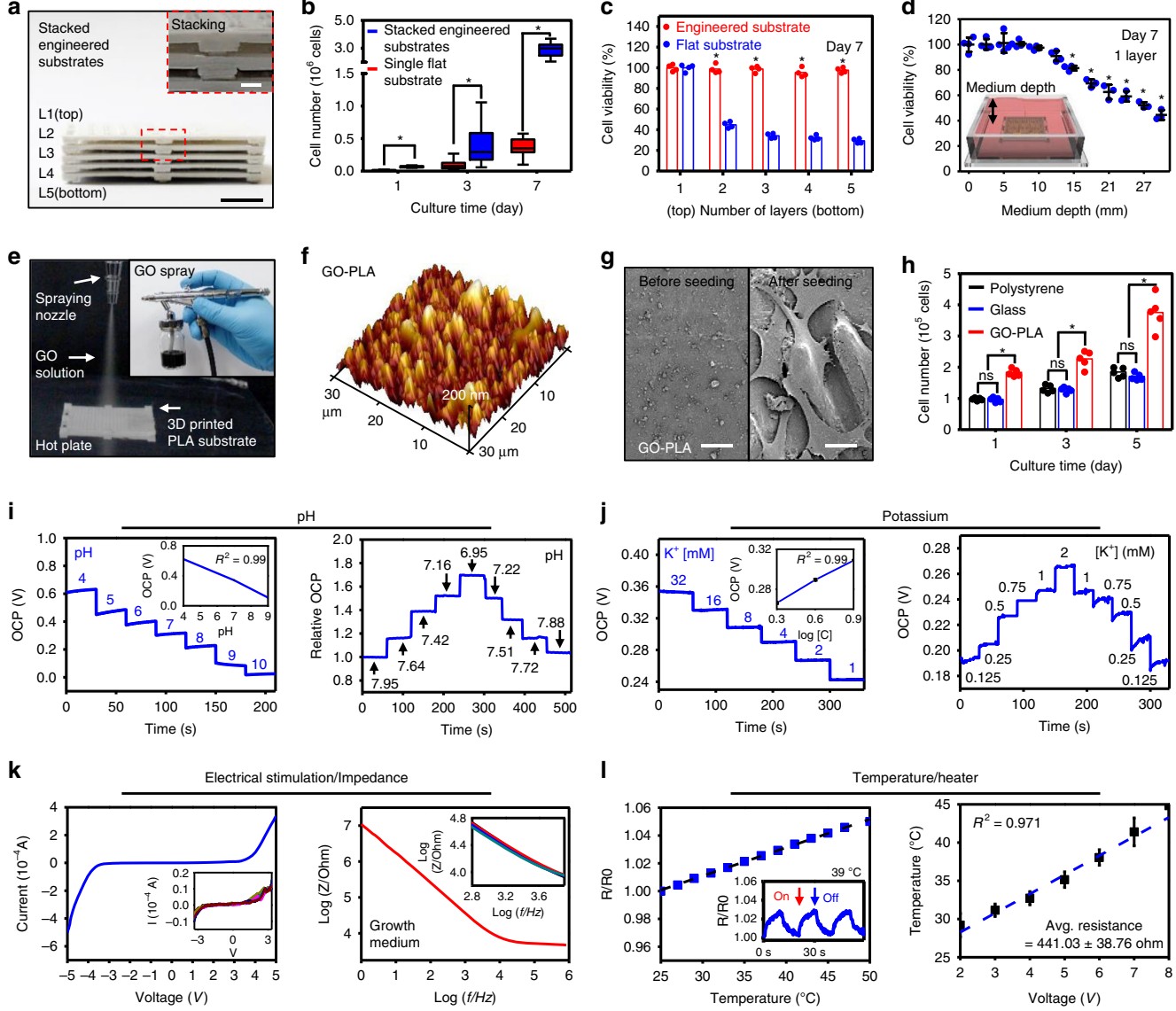

**Fig. 2** Characterization of cell culture substrate and ultrathin sensors and actuators. **a** Photograph of stacked 3D-printed engineered substrates (side view, scale bar: 1 cm) and magnified view of the assembly region (inset, scale bar: 2 mm). **b** Plot comparing cell attachment on a stack of engineered substrates and on a single flat substrate. ($n = 5$, Box: median; 25th to 75th percentiles, Whisker: min to max, *$P < 0.001$ with two tailed unpaired $t$ test). **c** Plot showing cell viability in each layer of the five-layer stack of engineered substrates and flat substrates. The number of the layers descends from the top layer. ($n = 4$, mean, *$P < 0.001$ versus flat substrate, ANOVA with Bonferroni's post-test). **d** Plot showing the viability of cells based on the depth of culture medium ($n = 3$, mean ± s.d., *$P < 0.001$ compared to control (depth = 0 mm), ANOVA). **e** Image showing GO spray-coating onto the 3D printed PLA substrate. Inset shows a spray containing GO. **f** AFM image of the GO-coated PLA substrate. **g** SEM images of the GO-PLA substrate before and after cell seeding. Scale bar: 20 μm. **h** Plot showing the number of cells cultured on polystyrene, glass, and GO-coated PLA substrates ($n = 4$, mean, *$P < 0.001$, ANOVA with Bonferroni's post-test, ns, not significant). **i–l** Characterization of sensors and stimulators. **i** The graphs show the open-circuit potential (OCP) measurements from pH sensor at the pH range of 4–10 (left) and in subtle changes between pH 6.95 and 7.95 compared to a commercial pH sensor (numbers on each step) (right). **j** The graphs show OCP measurements from K+ sensor at concentrations between 1 and 32 mM (numbers on each step) (left) and in subtle changes between 0.125 and 2 mM (right). The insets in **i**, **j** show calibration curves for both ion sensors ($n = 5$). **k** Current–voltage plot of electrical stimulator (left) and log–log plot of the impedance measurements at different frequencies (right). Insets show reproducibility ($n = 16$ for left and $n = 4$ for right). **l** Graphs showing the resistance of temperature sensor changes with temperature (left) and thermal stimulator response to sudden temperature variations (inset). Graph showing the relationship between heat and applied voltage ($n = 16$) (right)

(MUX). The electrical current that passed through the cells are consecutively converted to voltage and measured for impedance calculation (Fig. 3b). The circuits for impedance sensing also can be used for electrical stimulation, for which alternating electrical pulses generated by the DAC are injected into the MUX via a bypass switch to detour the low-pass filter (LPF) (Fig. 3c). Separately regulated voltages are selectively applied to the desired heaters for thermal stimulation (Fig. 3d). Open-circuit voltages

(OCVs) at terminals of pH and K+ sensors are transferred to measuring circuitries through voltage buffer circuits (Fig. 3e). A built-in Bluetooth module (BT) transfers all collected data to external devices and receives commands through a serial port for wireless control.

The frequency of impedance measurement can be modulated from 10 to 10,000 Hz in the embedded software (Fig. 3f, g). The alternate current at low frequency (e.g., range of $10^2$–$10^4$ Hz)

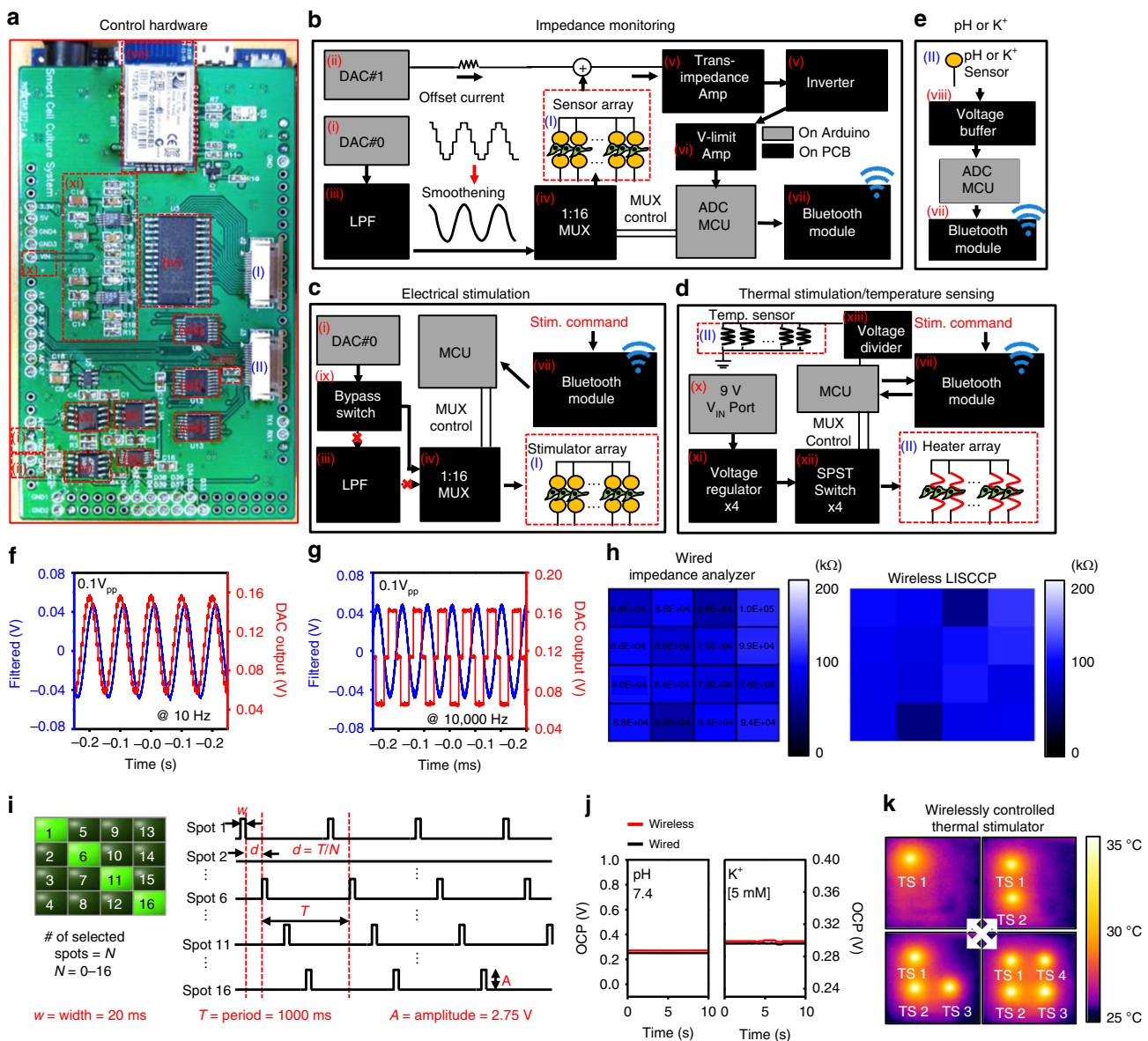

**Fig. 3** Design of a wireless system and control algorithm. **a** Photograph of the data acquisition hardware. **b**–**e** Control flow chart of the impedance sensing (**b**), electrical stimulation (**c**), thermal stimulation and temperature sensing (**d**), and pH and K+ sensing (**e**). **f, g** Plots showing the DAC output for impedance measurement (red) and filtered signals (blue) of the adjustable frequency range (10–10,000 Hz). **h** Comparison of the impedance measurement between wired (left) and wireless (right; screenshot image of the software) recordings. **i** Plot of electrical stimulations controlled by the user interface software to specify the site of stimulations. **j** Plots to compare wired and wireless measurements of pH and K+ concentrations. **k** Infrared camera images of the heat generated and controlled by the thermal stimulators (TS) wirelessly. The number of TS to turn on can be controlled

flows through the paracellular space of the cells[20]. To obtain sensitive detection on changes in the cell morphology such as differentiation of C2C12, we set the frequency as 200 Hz for all wireless impedance measurements. We compared wirelessly measured impedance with measurement of a wired impedance analyzer and found no significant difference (Fig. 3h). Users can select specific location for electrical stimulation through the user interface (e.g., spots 1, 6, 11, and 16; Fig. 3i left). The installed software controls the delay between stimulating pulses to apply same stimulations to multiple locations (Fig. 3i right). The wirelessly measured pH and K+ changes are compatible with measurement by a wired system in the OCP mode using a commercial electrochemical analyzer (CH Instruments, Inc) (Fig. 3j). The LISCCP also controls heating locations wirelessly for selective thermal stimulations (Fig. 3k).

**Wireless monitoring of cells on single-layer platforms**. Figure 4 shows wireless monitoring of cultures of four types of cells in the single-layer cell culture platform (single-layer CCP; Supplementary Fig. 15). Human dermal fibroblasts (hDFB), human mesenchymal stem cells (hMSC), mouse skeletal muscle cells (C2C12 myoblasts), and HL-1 mouse cardiac muscle cells (HL-1) are cultured separately in single-layer CCPs. While the single-layer CCPs reside in an incubator, the cell proliferation and differentiation are wirelessly monitored and controlled in situ in real time by the interactive user interface on a single laptop (Fig. 4a). The user interfaces can be easily switched for each mode of monitoring and stimulation (Fig. 4b).

The monitoring data of impedance, temperature, pH, and K+ for four kinds of cells cultured for 18 days is presented in Fig. 4c. The temperature is maintained at 37 °C throughout the culture

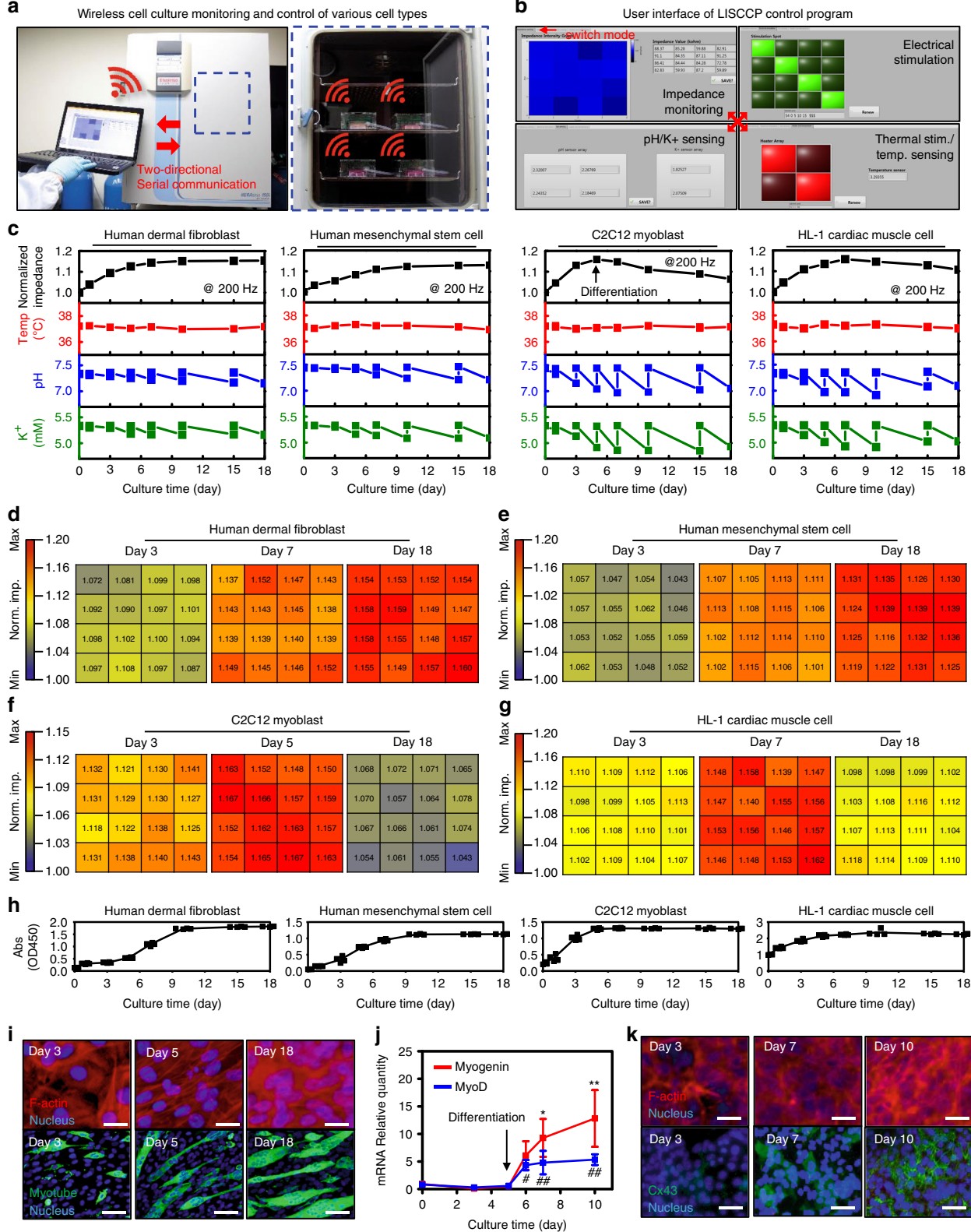

period. On the other hand, the pH and $K^+$ concentrations decrease as the cells proliferate and differentiate[21,22]. Although pH and $K^+$ concentrations fluctuate every time when the culture medium is changed, the decrease rates become steeper as the culture proceeds since the number of cells increases. This fluctuation can be reduced by continuous medium circulation, which will be shown in Fig. 5. The pH and $K^+$

sensors are calibrated every time when fresh culture medium is introduced.

The monitored impedance shows distinctive curves depending on the type, growth, and differentiation of the cultured cells[23,24]. The data from 16 impedance sensors in the array for hDFB, hMSC, C2C12, and HL-1 are shown as color mappings in Fig. 4d–g (see Supplementary Fig. 16 for more detailed data). The

**Fig. 4** Real-time monitoring of various cells cultured on a single-layer CCP. **a** Photographic image of four single-layer CCPs in an incubator, controlled wirelessly by a single laptop. **b** Screenshot images of the user interactive software for each mode of monitoring and stimulations: impedance sensing (top left), pH/$K^+$ sensing (bottom left), electrical stimulation (top right), and temperature sensing/thermal stimulation (bottom right). **c** Monitored data (impedance, temperature, pH, and $K^+$) for cultures of four types of cells (hDFB human dermal fibroblasts, hMSC human mesenchymal stem cell, C2C12 mouse myoblast, HL-1 mouse cardiac muscle cell; mean). In the C2C12 culture, the growth medium was changed to the differentiation medium on day 5 (arrow). **d–g** Impedance mappings of hDFB (**d**), hMSC (**e**), C2C12 (**f**), and HL-1 (**g**). The blue color indicates the lowest impedance, whereas red color indicates the impedance in maximum. **h** Growths of four types of cells as determined by the WST-8 assay. The absorbance is increased as the number of cells is increased. **i** Fluorescent microscopic images of F-actin-stained C2C12 cells during proliferation (top, scale bar: 10 μm) and myosin heavy chain staining of C2C12 cells showing myotube formation of the cells during differentiation (bottom, scale bar: 50 μm). **j** Expression of the muscle-specific gene for *myogenin* (red line) and *MyoD* (blue line) in C2C12 cells ($n = 3$, mean ± s.d., *myogenin* compared to day 0, *$P < 0.05$, **$P < 0.01$; *MyoD* compared to day 0 #$P < 0.05$, ##$P < 0.01$, ANOVA). **k** Fluorescent microscopic images of F-actinstained HL-1 during proliferation (top, scale bar: 40 μm) and connexin 43 (Cx43) expression of the cells during differentiation (bottom, scale bar: 30 μm)

impedance steadily increases (Fig. 4c–g) during cell proliferation for all types of cells (Fig. 4h) because the cells cover the surface of electrodes, acting as insulators to impede the path of electrical current[25]. The impedance of hDFB and undifferentiated hMSC continuously increases and saturates at the late period of the cell culture[23] (Fig. 4c–e).

On the other hand, C2C12 and HL-1 show an impedance tendency different from that of hDFB and undifferentiated hMSC (Fig. 4c, f, g). C2C12 proliferates until 100% confluency (Fig. 4h) that is shown as the highest peak of the impedance curve (Fig. 4c). Then the impedance decreases as C2C12 differentiates in the differentiation medium. In skeletal muscle cell differentiation, cells transform into fused, elongated, and multinucleated cells (i.e., myotubes), and the cell area covering electrodes is decreased[26]. Thus the impedance decreases in the myogenic differentiation stage of C2C12 as the myotube formation occurs[26]. Immunostaining of C2C12 shows that the F-actin of cell bodies covers the electrodes as the cells proliferate initially, while myotubes and multi-nucleus cells are observed during differentiation (Fig. 4i). The muscle-specific gene markers such as *myogenin* and *MyoD* are also upregulated after the induction of differentiation (Fig. 4j). Meanwhile, the impedance of HL-1 (Fig. 4c, g) also increases for days and then decreases, which is similar to that of C2C12. After the impedance peaks at maximal cell concentration of HL-1, the impedance decreases (Fig. 4c, g, h) due to increased cell-to-cell electrical coupling via increased cell–cell contact at high cell concentration. The increased expression of connexin 43 (Cx43), a gap junction protein of cardiomyocytes in the differentiation stage[27], is shown in Fig. 4k. More connections within adjacent cells increases the electrical pathway, which slightly decreases intercellular impedance. The increased expression of myotube for C2C12 and Cx43 for HL-1 are quantitatively described in Supplementary Fig. 17. Collectively, the impedance curves measured by the single-layer CCP are consistent with the biological analysis.

**Wireless monitoring and stimulation in 3D multi-layer array.**
Figure 5 shows real-time, wireless, 3D multi-layer array monitoring and in situ local stimulation in the large-scale cell culture of C2C12 in a five-layer LISCCP. The 3D impedance (Fig. 5a) and pH (Fig. 5b) mappings are shown for proliferation and differentiation of C2C12 at days 5, 7, and 18. The color maps of the impedance are largely homogeneous throughout five layers at each time point. Initially, the impedance increases as cells proliferate. It reaches a maximum value on day 7 and then decreases after the differentiation (Fig. 5a). On the other hand, the pH monitoring shows that pH at day 18 is much lower than pH at day 5 because pH decreases faster when the cell number is higher (Fig. 5b). The pH at the bottom layer is more acidic than that at the top layer due to higher production of lactic acid at the bottom

layer where diffusion of dissolved oxygen is limited[28]. The $K^+$ concentration are also monitored, and all raw data are shown in Supplementary Fig. 18. Co-plots of the impedance monitoring from each sensor for each layer are shown in Supplementary Fig. 19.

To improve mass transfer (e.g., oxygen) in the multilayer cell culture, culture medium can be circulated using a peristaltic pump[29] (Fig. 5c). An image of the five-layer LISCCP integrated with the peristaltic pump via inlet and outlet tubes in an incubator is shown in Supplementary Fig. 20a. Finite-element method analysis shows that the dissolved oxygen concentration is substantially homogeneous throughout all culture layers after continuous culture medium circulation (Supplementary Fig. 20b) compared to the case without circulation (Supplementary Fig. 8b). Accordingly, pH fluctuation with medium circulation is smaller than that without circulation (Fig. 5d) due to the improved mass transfer. Culture medium circulation[30] improves the mass transfer (Supplementary Fig. 20c) and enables the number of culture layers in LISCCP to be increased up to 25 layers without diminishing cell viability significantly (Fig. 5e, f).

LISCCP can promote cellular proliferation and differentiation via electrical/thermal stimulations in a wireless manner. Electrical stimulation alters the resting transmembrane potential, which upregulates the expression of growth factors[31]. Thermal stimulation induces mitochondrial biogenesis and enhance AMP-activated protein kinase activity[32]. Both electrical and thermal stimulations (ES and TS, respectively) are applied to C2C12 myoblast culture on a single-layer CCP according to the timetable and parameters[31–34] shown in Supplementary Fig. 21a. The impedance of the stimulated cells increases and later decreases faster compared to the cells without stimulations (Fig. 5g). This implies that the stimulated cells proliferate and differentiate faster. The enhanced cell proliferation and differentiation by the stimulations are confirmed by muscle-specific marker gene quantification (*myogenin* and *MyoD*, Fig. 5h) and by the fluorescence analysis (F-actin staining on day 5 and myotube staining on day 10, Fig. 5i).

Furthermore, cell proliferation and differentiation can be locally stimulated by applying ES and TS to a specific layer of LISCCP (e.g., third layer in Fig. 5j–n) or even at specific local points (Supplementary Fig. 21c–g; round dots represent localized ES and TS sites). For example, ES and TS on the third layer (L3) results in higher initial impedance due to faster cell proliferation (red color map in Fig. 5j and blue line in Fig. 5k) and lower late impedance due to faster differentiation (blue color map at day 18 in Fig. 5j and blue line in Fig. 5k) on the third layer. The number of cells on the stimulated layer on day 5 is highest among all layers (Fig. 5l), which demonstrates the enhanced proliferation by the stimulation. The enhanced differentiation at L3 is confirmed by faster upregulation of a muscle-specific marker gene (*MyoD*) (Fig. 5m and Supplementary Fig. 21b). Cells at L2 and L4 are also

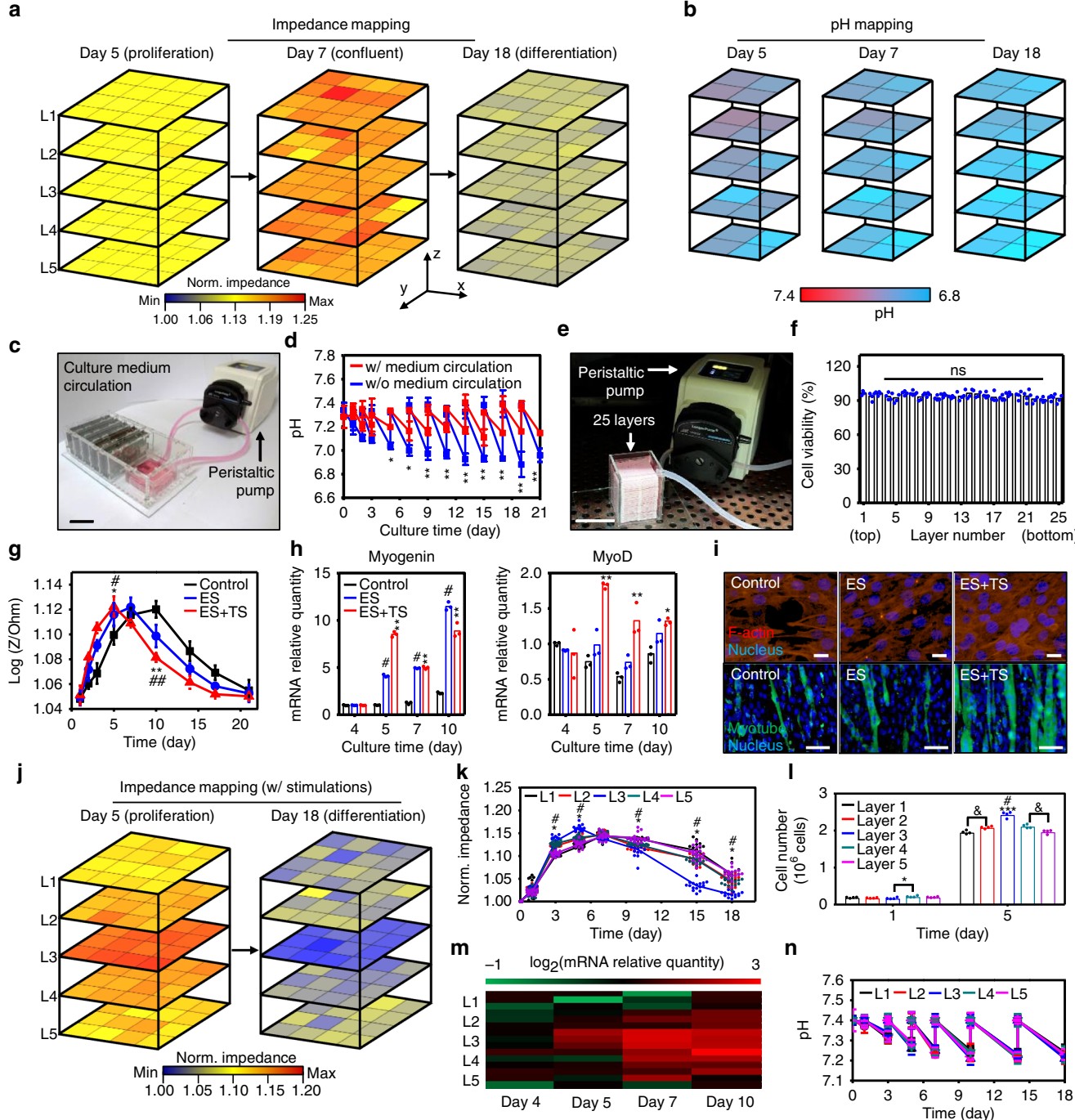

**Fig. 5** 3D array monitoring and stimulation control of C2C12 culture with LISCCP. **a** Impedance mapping of C2C12 cells cultured on the five-layered LISCCP without stimulations. Confluence appears at day 7 (red) before the differentiation proceeds. **b** Corresponding 3D pH mapping to **a**. **c** Photograph of LISCCP integrated with a peristaltic pump for culture medium circulation. Scale bar: 5 cm. **d** Profiles of pH with and without circulation of culture medium ($n = 4$, mean ± s.d., *$P < 0.01$, **$P < 0.001$, ANOVA). **e** Photograph of 25 culture layers with a culture medium circulation unit. Scale bar: 5 cm. **f** Cell viability at each layer in the 25-layer LISCCP ($n = 5$, ns, no significant; ANOVA). **g** Impedance monitoring for C2C12 cells cultured under ES, TS, or ES+TS. No stimulation was applied for the control ($n = 4$, mean ± s.d., ES+TS compared to control, *$P < 0.01$, **$P < 0.001$; ES compared to control, #$P < 0.01$, ##$P < 0.001$, ANOVA). **h** Expression of the muscle gene markers, *myogenin* (left) and *MyoD* (right), during the differentiation period ($n = 3$, mean ± s.d., ES+TS compared to control, *$P < 0.01$, **$P < 0.001$; ES compared to control, #$P < 0.001$, ANOVA with Bonferroni's post-test). **i** Fluorescent analysis of C2C12 cells cultured without (control) and with stimulations (ES, TS, ES+TS). F-actin is stained red on day 5, myotube in green on day 10, and nucleus in blue. Scale bars for F-actin: 10 μm and for myotube: 50 μm. **j** Impedance mapping of a five-layered LISCCP when ES and TS are applied only on the third layer. **k** Impedance profiles of C2C12 culture on five layers with ES and TS (*$P < 0.001$, L3 compared to L1; #$P < 0.001$, L2 compared to L1). **l** Cell growth profiles showing increased cell growth on the stimulated layer ($n = 4$, *$P < 0.05$; ***$P < 0.001$ versus layer 1 and layer 5; #$P < 0.001$ versus layer 2 and layer 4, &$P < 0.05$, ANOVA). **m** Myogenic gene expression in C2C12 during differentiation ($n = 3$). **n** Profiles of pH on five layers ($n = 4$, mean ± s.d.)

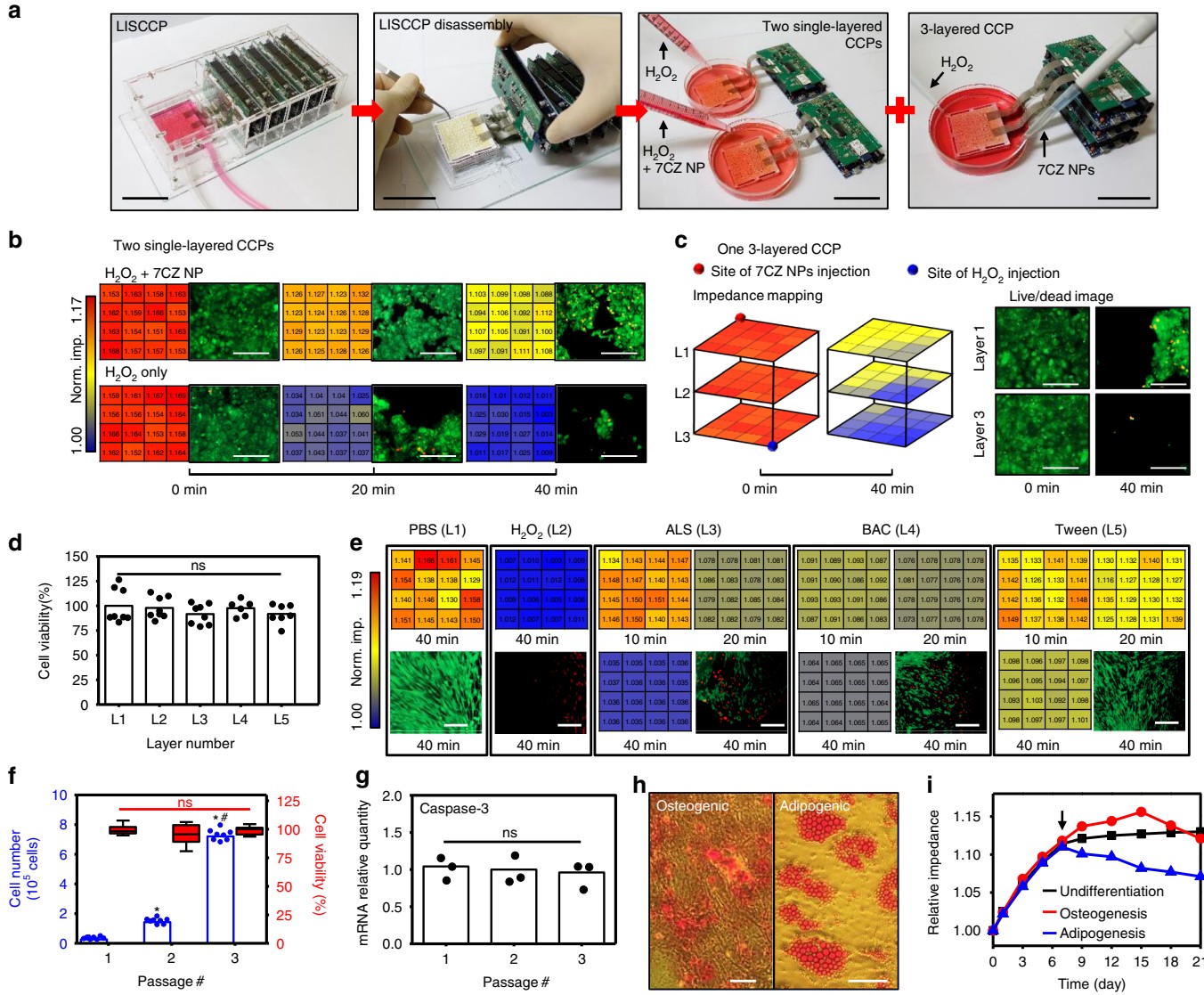

**Fig. 6** Applications of multilayerd LISCCPs. **a** Photographs showing a multilayered LISCCP used for in vitro drug toxicity testing of drugs and cosmetics. LISCCP is divided into two single-layer CCPs and one triple-layered CCP to conduct three different tests. Scale bar: 5 cm. **b** Testing to evaluate the effects of 7CZ NP-mediated ROS scavenging on HL-1 using single-layer CCPs. The impedance mappings of HL-1 culture along with its corresponding live (green)/dead (red) cell images are shown for the treatment of $H_2O_2$+7CZ NPs (top) and $H_2O_2$ only (bottom). Scale bar: 500 μm. **c** Triple-layered impedance mapping of HL-1 treated with $H_2O_2$ at one corner and 7CZ NPs at the opposite corner (left) and their live/dead cell images (right). Scale bar: 500 μm. **d** Viability of hDFB cultured on each layer of LISCCP at $t = 0$ ($n = 8$, mean ± s.d., ns, not significant, ANOVA). **e** Impedance mappings and live/dead cell images of hDFB treated with phosphate-buffered saline (PBS), $H_2O_2$, or one of the surfactants (ALS ammonium lauryl sulfate, BAC benzalkonium chloride, Tween TWEEN 60). Scale bar: 100 μm. **f** Viability and number of hMSC cultured on LISCCP for three passages (cell number, $n = 8$, *$P < 0.001$ versus passage 1, #$P < 0.01$ versus passage 2, ANOVA with Bonferroni's post-test; cell viability, $n = 4$, Box: median; 25th to 75th percentiles, Whisker: min to max, ns, not significant, ANOVA). Large expansion of cells shows the utilization of LISCCP in large-scale cell manufacturing for cell therapy. **g** Caspase-3 gene expression of hMSC cultured on LISCCP for three passages ($n = 3$, ns, not significant, ANOVA with Bonferroni's post-test). **h** Staining images of **h**MSC differentiated into osteogenic (left, Alizarin red O staining) and adipogenic (right, Oil red O staining) lineage cells. Red indicates osteogenic or adipogenic lineage cells. Scale bar: 50 μm. **i** Impedance monitoring of hMSC cultures that were induced to undergo osteogenic and adipogenic differentiation by medium change on day 7 (arrow) when the cells were 90% confluent

stimulated (Fig. 5m), which may be due to propagation of heat generated from the thermal stimulations and paracrine-soluble factors secreted from earlier differentiated muscle cells at L3 to the surrounding layers. The pH profile, however, shows no difference among all five layers regardless of the stimulation (Fig. 5n), which may be due to the continuous medium circulation.

**Industrial application potential of LISCCP.** Figure 6 presents various applications of LISCCP, such as screening of drug

candidates (Fig. 6a–c), toxicity testing of cosmetic ingredients (Fig. 6d, e), and large-scale production of therapeutic cells (Fig. 6f–i). For in vitro screening and testing, each layer of LISCCP can be separated for multiple testing of chemicals[35]. For example, two single-layered and one three-layered CCPs can be prepared from one LISCCP (Fig. 6a).

As an example, LISCCP can be used to test the capability of ceria-zirconia nanoparticles (7CZ NPs) to scavenge reactive oxygen species (ROS)[36] in the cardiac disease model in vitro using HL-1 culture. ROS are regarded as a major cause of

inflammatory diseases such as myocardial infarction and stroke[37]. The 7CZ NP synthesis and the dosage optimization process for ROS scavenging are shown in Supplementary Fig. 22a–e. Impedance of HL-1 cultures is recorded in two different single-layer CCPs after treatment with either $H_2O_2$ (as ROS; 10 mM) only or $H_2O_2$ (10 mM)+7CZ NPs (0.2 mM). The impedance mappings indicate that 7CZ NPs prevented cellular damage caused by ROS, which is confirmed by the live/dead assay (Fig. 6b). The treatment with $H_2O_2$ only leads to nearly complete cell death (after 40 min; Fig. 6b, bottom), which is shown as minimum impedance (blue; Fig. 6b bottom). Figure 6c illustrates 3D impedance monitoring of HL-1 cultured in a three-layered CCP after treating the cells with $H_2O_2$ (10 mM; blue dot) and 7CZ NPs (0.2 mM; red dot) at different corner points. The impedance at the $H_2O_2$ injection site is very low due to cell death and the subsequent detachment of dead cells (Fig. 6c left). The detailed impedance mappings are shown in Supplementary Fig. 22f. Meanwhile, 7CZ NPs scavenge ROS, reducing the cellular death. This observation is confirmed by data of live/dead assays (Fig. 6c right).

The potential irritation of cosmetic ingredients has been evaluated using animal experiments, but demands for in vitro screening are increased with the rise of animal rights. For in vitro skin toxicity testing, hDFB is cultured in a five-layered LISCCP, which is disassembled into five single-layer CCPs for five different testing scenarios. Three types of surfactants, ammonium lauryl sulfate (ALS), benzalkonium chloride (BAC), and Tween 60 (Tween), are chosen as potential ingredients that can affect cell viability[38]. The initial high viability of hDFB is confirmed (Fig. 6d). The high cellular viability is maintained with phosphate-buffered saline (PBS), while most of cells are completely dead and detached from the culture surface with 100 mM $H_2O_2$ (L1, L2; Fig. 6e). PBS and $H_2O_2$ are used for positive and negative controls, respectively. The impedance observations are confirmed by the live/dead assay (bottom row of Fig. 6e). With 0.1 mM ALS and 0.01 mM BAC, many cells are detached from the substrate and dead after 40 min, showing low impedance. Since ALS and BAC are anionic and cationic surfactants, respectively, cell adhesion to the culture surface is weakened[39]. On the other hand, treatment with 5% w/w Tween (L5; Fig. 6e), a nonionic surfactant, shows minimal cytotoxicity even after 40 min. The impedance color map and their cellular viabilities are shown in Supplementary Fig. 23.

Additionally, LISCCP can be used for large-scale production of therapeutic cells. One of the most commonly used cell types for cell therapy is hMSC[40,41]. Large expansion (27 folds) of hMSC is demonstrated by three passages in LISCCP (Fig. 6f left). The high cell viability is maintained (Fig. 6f right) and the expression of caspase-3, a cell-apoptosis indicator[42], does not increase during the repeated cultures in LISCCP (Fig. 6g). The hMSC expanded in LISCCP can be encapsulated in hydrogel (Matrigel)[43] for subsequent injection (Supplementary Fig. 24). These results show that LISCCP can produce a large number of viable hMSC for cell therapy, which preserve differentiation capacity after the cell expansion. Moreover, hMSC are able to differentiate into specific types of cells for cell transplantation therapy[44] under real-time monitoring in LISCCP (Fig. 6h, i). For example, the impedance increased after induction of osteogenic differentiation, whereas it decreases after induction of adipogenic differentiation[23].

## Discussion

Here we develop a multifunctional LISCCP for the digital mass culture of anchorage-dependent cells through multi-scale integration of ultrathin sensor and actuator arrays. Various sensors and actuators are fabricated as arrays, the fabricated ultrathin

arrays are integrated through the transfer printing methods, and CCPs are stacked to form a LISCCP. The wireless system enables integration of multiple LISCCPs across multiple incubators and digital monitoring and local control of numerous culture layers, making the large-scale cell culture more efficient.

The integrated sensor arrays on each substrate layer, which is stacked into multiple layers in a chamber of LISCCP, allow thorough 3D monitoring of the electrophysiological and environmental conditions of cultured cells. Furthermore, the arrays of electrical and thermal stimulators can locally control cellular proliferation and differentiation. Since the PLA substrate layers are 3D printed and designed like a Lego® block, the assembly of each layer as well as the disassembly for the separate use of each layer is facile. The perforations on the substrate allow for diffusion of dissolved oxygen and nutrients throughout the multiple layers. For large-scale cell cultures, the number of culture substrate layers in LISCCP can be increased up to 25 layers with the aid of culture medium circulation system. The peristaltic pump of this system can reside outside of an incubator, while the pump is connected to LISCCP via the inlet and outlet tubes.

Although the culture medium circulation system allows the cell culture of up to 25 layers, integration of automated medium perfusion facility in the culture system may increase the culture substrate number to >25 layers. The integration of the culture medium perfusion facility can also prevent fluctuation of pH and ion concentrations within the culture medium. Meanwhile, the miniaturization of the wireless system in LISCCP can further increase the space efficiency so that more LISCCPs can be placed in an incubator. In the future study, the current Arduino-based wireless system can be replaced by the customized wireless chip, which will decrease the overall system size further.

LISCCP has several advantages over conventional methods for in vitro drug screening and toxicity testing. Conventional cytotoxicity tests with biochemical assays such as water-soluble tetrazolium salt-8 assay and fluorescent live/dead cell assay require incubation time, reagents, and sacrifice of cell samples for analysis. In contrast, LISCCP monitoring allows for real-time toxicity analysis without reagents and sacrifice of cell samples. Therefore, the state of cells such as cell viability can be monitored continuously in real time by LISCCP. Moreover, a large number of drug candidates can be automatically applied to each platform of LISCCP through silicon tubes, if the multi-channel pumping facilities are installed.

Furthermore, LISCCP can be an appropriate system for clinical-scale production of therapeutic cells. LISCCP allows therapeutic cells to be expanded into a large number, while maintaining high viability and differentiation potency. Without conventional staining methods that accompany sacrifice of cells, LISCCP allows estimation of the completion time period for differentiation of cultured cells. Also, LISCCP does not need to be taken outside of the incubator for monitoring/stimulation of the cells or for change of culture medium. The entire LISCCP system is automated, digitally monitored, and locally controlled, and thereby labor-efficient and highly productive, which will transform conventional cell cultures into digital mass cultures.

## Methods

**Fabrication of sensors and stimulators**. The electrode arrays were fabricated on Ni-deposited silicon wafers before being transferred onto the engineered PLA substrate. The array layout, thickness information, and illustrations of fabrication process are provided in Supplementary Figs. 3–5, respectively. Briefly, 20 nm of Ni was thermally deposited onto silicon wafers. Then SU8 (~1.5 μm; SU8-2, Microchem) along with PI (~1.3 μm; polyamic acid, Sigma-Aldrich) were coated and patterned by photolithography. On top of these polymer layers, 7 nm/70 nm of Cr/Au were thermally deposited and patterned into temperature sensors and encapsulated with SU8. Next, 7 nm/150 nm of Cr/Au were deposited and patterned into a thermal stimulator and encapsulated with SU8. For pH and $K^+$ sensors, 40 nm/150 nm of Cr/Au were

patterned and encapsulated with patterned SU8. Next, 7 nm/100 nm of Cr/Au were patterned into impedance sensors/electrical stimulators and encapsulated with PI. For complete encapsulation to prevent any current leakage to the culture medium, 200 nm of $SiO_2$ and 20 nm of $Al_2O_3$ were deposited using sputter and atomic layer deposition, respectively, and patterned to create vertical interconnect access (VIA). The metal oxide and thin polymer layers were encapsulated with SU8. Photoresist was patterned to cover only the VIA of each sensor and was later removed by the lift-off process after spray-coating the entire surface with GO. The working electrodes for the pH and $K^+$ sensors were connected to a PCB by anisotropic conductive film for electrodeposition. After selective functionalization, the sacrificial layer of Ni was wet etched, and the electrode arrays were transfer printed onto the engineered PLA substrate. The substrate was treated with chloroform solution before the arrays being transfer printed and then annealed at 55 °C for 1 h. The SEM images of the device cross-sections in Supplementary Fig. 4 are obtained by FE-SEM 7800F Prime installed at the National Center for Inter-university Research Facilities at Seoul National University

**Selective functionalization of pH and $K^+$ sensors**. Selective functionalization was performed by electrodeposition using the three-electrode method with the reference electrode (commercial Ag/AgCl electrode), Pt counter electrode, and the fabricated working electrode via electrochemical analyzer (CH Instrument, Inc). Polyaniline (PANi) was electrodeposited onto the sensing component of the pH sensor. An aqueous solution of 0.1 M aniline (Sigma-Aldrich) in 1 M HCl was prepared. The fabricated Au working electrode was submerged into the solution, and the potential was swept from −0.2 to 1 V versus a commercial Ag/AgCl electrode for 60 segments at a scan rate of 0.1 V s$^{-1}$. Here poly(3,4-ethylenedioxythiophene) polystyrene sulfonate was used as the ion-electron transducer of $K^+$ sensor. Then an aqueous solution of 0.01 M 3,4-ethylenedioxythiphene (Sigma-Aldrich) and 0.1 M sodium polystyrene sulfonate (Sigma-Aldrich) was prepared. The working electrode was submerged into the solution and the chronopotentio-metric electrodeposition was conducted at 0.02 mA anodic current (potential versus commercial Ag/AgCl electrode) for 200 s. The common reference electrode was fabricated with screen printing of Ag/AgCl paste (Gwent Electronic Materials) onto an Au electrode that was subsequently coated with polyvinyl butyl resin (PVB) (BUTVAR B-98, Sigma-Aldrich).

**Preparation of $K^+$-selective membranes**. The $K^+$-selective membrane cocktail[45,46] was prepared by valinomycin (2% w/w), sodium tetrakis[3,5-bis(tri-fluoromethyl)phenyl] borate (0.5% w/w), high-molecular-weight polyvinyl chloride (32% w/w), and bis(2-ethylhexyl) sebacate (66% w/w). Then 100 mg of the membrane cocktail was dissolved in 350 μL of tetrahydrofuran. The solution for the PVB reference electrode was prepared by dissolving 79.1 mg PVB and 50 mg NaCl into 1 mL methanol. All materials used for the $K^+$-selective membrane were purchased from Sigma-Aldrich.

**Design of 3D engineered substrate**. The engineered substrate was three-dimensionally printed with PLA filament using a 3D printer (Rokit Invivo, Rokit). The substrate (60 mm × 60 mm × 1 mm) contained a total of 36 square perforations (each 1 mm × 1 mm), 176 protruding hemispheres (each 1 mm in diameter), and 7 holes and pillars at the edges.

**GO spray coating**. A solution of GO was prepared by adding 160 mg of GO powder (Sigma-Aldrich) in 100 mL of ethanol and treated with ultra-sonification. The electrode arrays on the silicon wafer and 3D-printed PLA substrate were placed on a hot plate that was preheated to 60 °C. For each single layer, 4 mL of the GO solution was spray-coated from a distance of 20 cm using an airbrush at a pressure of 40 psi (HP-BCP, ANEST IWATA-MEDEA). The substrates were subsequently placed on a hot plate (60 °C) for 5 min to remove any remaining solvent.

**Culture medium circulation system**. The custom-made acrylic well (6.5 cm × 6.5 cm) contained two holes (inlet and outlet) along one side for culture medium passages connected via silicon tubes. The silicon tubes ($D = 20$ mm) connected together via a connector for easy exchange of culture medium were inserted into a peristaltic pump (Longer) to circulate the culture medium at a flow rate of 12 mL min$^{-1}$. The platforms resided inside of the custom-made acrylic well, and the bottom edges of the well were glued to the substrate glass with a silicone glue gun.

**Wireless hardware operation**. The flow chart in Fig. 3b shows the procedures used for impedance measurements. The alternating current signal, which was generated by a DAC (i) and filtered by an LPF (iii), was passed consecutively through each impedance sensor via a 1:16 MUX (iv). To make the signal fit within the measurable range, another DAC (ii) provided an offset current before signal amplification and inversion (v). The analog–digital converter (ADC) in the MCU measured the amplified signals, which were limited by a voltage-limiting amplifier (vi) for circuit protection. The measured data was transmitted to a BT (vii) for wireless data transmission to an external monitoring device. For electrical stimulation (Fig. 3c), the impedance measurement circuits were utilized with additional bypass switches (ix) that converted the impedance measurement mode to electrical stimulation mode and vice versa. After the user sent the commands and

stimulation location information via BT (vii), the installed software configured the bypass switches accordingly. Figure 3d illustrates the temperature-sensing and heater-control procedures. Because each heater had a different resistance, the different actuation voltages were regulated by programmable voltage regulators (xi). After the user sent the commands and stimulation location information via BT (vii), the installed software configured the switches (xii) to turn on the appropriate heaters, and the MCU measured the temperature using a voltage dividing circuit (viii). The measured temperature was also transmitted through BT (vii). Because of the high impedance of the terminals of the pH and $K^+$ ion sensors, voltage buffers were required in front of the ADC for accurate measurement of the OCVs (Fig. 3e).

**Characterizations of sensors and stimulators**. The electrochemical sensors were characterized to determine the calibration curves for their selectivity, sensitivity, and reproducibility. For the pH and $K^+$ sensors, a two-electrode system in which Ag/AgCl electrodes served as both the reference and counter electrode was used to simplify the circuits. The pH potential was measured in the time-basis while the sensor was submerged in different pH solutions. The calibration curve for $K^+$ was measured using the same set-up as that for pH. For measuring pH and $K^+$, the sensors were calibrated each time fresh cell culture medium was introduced. The pH and $K^+$ were calibrated to pH 7.4 and 5.3 mM $K^+$ concentration, respectively. To determine the calibration curve for impedance sensing, the impedance between two Au electrodes (working and reference/counter electrode) was measured while sweeping currents of frequencies between 1 and 1,000,000 Hz. The electrical stimulation was applied through the same electrodes as the impedance sensor, and the voltage was measured from a commercial Pt electrode in PBS. The resistance of the temperature sensor was measured by placing the sensor on a hot plate with increasing temperature from room temperature to 50 °C. The temperature sensor was placed immediately underneath the thermal stimulator. The heat produced from the thermal stimulator was measured continuously while the heat was produced periodically. Also, the infrared image of the heat generated from the heater is captured using thermography camera (FLIR System). All sensors were characterized with wired electrochemical analyzer (CH Instrument, Inc) before operated with the wireless hardware.

**Cell culture**. C2C12 myoblasts (C2C12; CRL-1772, ATCC), hDFB (PCS-201-012, ATCC), HL-1 (SCC065, Merck), and hMSC (PT-2501, Lonza) were used for the experiments. The growth medium for C2C12 was Dulbecco's modified Eagle's medium (DMEM, Gibco) supplemented with 10% (v/v) fetal bovine serum (FBS, Gibco), 100 units mL$^{-1}$ penicillin (Gibco), and 100 μg mL$^{-1}$ streptomycin (Gibco). The differentiation medium for C2C12 was DMEM supplemented with 5% horse serum (Gibco), 100 units mL$^{-1}$ penicillin, and 100 μg mL$^{-1}$ streptomycin. To monitor both proliferation and differentiation of C2C12 on the single-layer platform, we cultured the cells with differentiation medium. For C2C12 culture on five-layered LISCCPs, no differentiation medium was treated to see only electrical/thermal stimulation effect on the enhanced proliferation and differentiation of C2C12 cells. The growth medium for hDFB was DMEM supplemented with 10% FBS, 100 units mL$^{-1}$ penicillin, and 100 μg mL$^{-1}$ streptomycin. The growth medium for HL-1 was Claycomb medium supplemented with 10% FBS, 100 units mL$^{-1}$ penicillin, 100 μg mL$^{-1}$ streptomycin, 0.1 mM norepinephrine, and 2 mM L-glutamine. Finally, hMSC was cultured in mesenchymal stem cell growth medium (Lonza) and differentiated into osteocytes and adipocytes using the StemPro Differentiation Kits (Thermo Fisher Scientific). To evaluate cell counts in Figs. 2 and 6, cells were cultured on the substrates (2 cm × 2 cm). C2C12, hDFB, hMSC, and HL-1 were seeded at a density of $1 \times 10^4$ cells cm$^{-2}$, 5000 cells cm$^{-2}$, $1 \times 10^4$ cells cm$^{-2}$, and $1.2 \times 10^4$ cells cm$^{-2}$, respectively.

**Analysis of cellular characteristics**. The number of cells was determined using hemocytometer after staining the cells with Trypan Blue (0.4%; Invitrogen). Cell viability was determined by using 3-(4,5-dimethylthiazol-2-yl)−2,5-diphenylte-trazolium bromide assay kit (Abcam) and also by counting viable cells after staining the cells with Trypan Blue. The cell proliferations in Fig. 4h were analyzed using cell counting kit-8 (Dojinjo) assay. To qualitatively evaluate cell viability, cells were stained using the LIVE/DEAD Viability/Cytotoxicity Kit (Thermo Fisher). The multiple substrate layers were dissembled into a single layers to conduct biological assays, such as cell counting, cell viability determination, and the Live and Dead assay.

The mRNA levels in cells were analyzed by quantitative polymerase chain reaction (qRT-PCR). The total RNA was extracted from the samples ($n = 3$) using TRIzol reagent (Invitrogen) and reverse-transcribed into cDNA (AccuPower® RT-PCR PreMix, Bioneer, Republic of Korea). The expression of *MyoD*, *myogenin*, and caspase3 (Supplementary Table 1) were evaluated using the StepOnePlus real-time PCR system (Applied Biosystems, USA) and the $2^{-\Delta\Delta Ct}$ method. The PCR consisted of 60 cycles of denaturing (95 °C, 10 s), annealing (60 °C, 15 s), and elongation (72 °C, 30 s). The housekeeping gene, *glyceraldehyde 3-phosphate dehydrogenase* (*GAPDH*), expression level was used for normalization of the target genes.

To analyze the protein marker expression of C2C12 and HL-1, the cells were stained immunochemically. To stain F-actin, the cells were fixed in 4% paraformaldehyde solution, permeabilized in cytoskeletal (CSK) buffer, and blocked in blocking buffer. Then the cells were treated with rhodamine phalloidin (Thermo Fisher) for 20 min at room temperature. After washing, the cells were

counterstained with 4,6-diamidino-2-phenylindole (DAPI; Thermo Fisher) and observed under a fluorescence microscope (Eclipse Ti, Nikon). C2C12 cells were fixed in 4% paraformaldehyde solution, permeabilized in CSK buffer (50 mM NaCl, 150 mM sucrose, 2 mM $MgCl_2$, 50 mM Tris-base (Sigma), Triton X-100 0.5% (Sigma)), and blocked in blocking buffer (1:10 dilution; Abcam). The cells were then incubated with anti-myosin heavy-chain antibodies (ab683, 1:100 dilution; Abcam), followed by incubation with Alexa Fluor 488 donkey anti-mouse IgG (ab150105, 1:100 dilution; Abcam) for 1 h at 37 °C. After washing, the cells were counterstained with DAPI (Thermo Fisher) and observed under a fluorescence microscope (Eclipse Ti, Nikon). HL-1 cells were fixed in 4% paraformaldehyde solution, permeabilized with PBS supplemented with 0.15% (v/v) Triton X-100 (Sigma) and 10% (v/v) goat serum (Gibco) for 2 h at room temperature, and blocked in blocking buffer. The samples were then reacted with Cx43 rabbit antibody (ab11370, 1:200, Abcam) overnight at 4 °C. After washing with PBS, the cells were reacted with fluorescein isothiocyanate-conjugated anti-rabbit secondary antibody (111-095-144, 1:50, Jackson ImmunoResearch Laboratories, USA) for 1 h at room temperature. The samples were mounted in DAPI (Vector Laboratories, USA) for nuclear staining. For the quantification of the F-actin-, myosin heavy chain-, and Cx43-positive area, the positive area was counted via the percentage relative to the total image area using the ImageJ software ($n = 4$).

To evaluate the viability of hDFB in surfactant assay, the cells were treated with surfactants (ALS solution (0.1 wt%; Sigma), BAC (0.1 wt%; Sigma), and Tween (5 wt%; Sigma)) and stained with the LIVE/DEAD Viability/Cytotoxicity Kit (Thermo Fisher). To evaluate hMSC differentiation into osteogenic and adipogenic lineage cells, the cells were counter stained with 2% Alizarin Red Stain (Lifeline Cell Technology) and Oil Red O Staining Kit (Lifeline Cell Technology)

**Synthesis of $Ce_{0.7}Zr_{0.3}O_2$ nanoparticles (7CZ NPs).** First, 0.36 mg of cerium (III) acetylacetonate hydrate (Sigma-Aldrich) and 0.14 mg of zirconium (IV) acetylacetonate hydrate (Sigma-Aldrich) were added to 15 mL of oleylamine (approximate C18-content of 80–90%; Acros Organics). The mixture was stirred for 15 min at 20 °C and heated to 80 °C at a heating rate of 2 °C min⁻¹. The mixture was aged at 80 °C for 1 day to obtain a dark brownish solution and cooled to 20 °C. The product was washed with acetone (100 mL; Samchun Chemicals) and collected by centrifugation. The collected 7CZ NPs were kept in chloroform (Samchun Chemicals) at a concentration of 10 mg mL⁻¹. To transfer the 7CZ NPs into water, 2 mL of the 7CZ NPs in chloroform (10 mg mL⁻¹) was mixed with 6 mL of 1,2-distearoyl-sn-glycero-3-phosphoethanolamine-N-[methoxy(polyethyleneglycol)−2000] (mPEG(2000)-PE; Avanti Polar Lipids) in the chloroform (10 mg mL⁻¹). Using both the rotary evaporator and vacuum oven, chloroform was completely removed to obtain fine dried PEGlyated-7CZ NPs. Then 5 mL of deionized (DI) water was added to the sample to make a transparent colloidal suspension. After sonicating for 10 min and filtration using a 0.4-μm filter, the excess mPEG(2000)-PE was removed by ultracentrifugation. The cleansed PEGylated-7CZ NPs were finally dispersed in DI water or PBS at the desired concentration.

**Statistics.** All statistical analysis was performed by using the GraphPad Prism software v5.03.

**Reporting summary.** Further information on research design is available in the Nature Research Reporting Summary linked to this article.

## Data availability
The data that support the findings of this study are available from the corresponding author upon request.

## Code availability
We used Arduino 1.8.3 and LabVIEW 2017 to operate our custom-made circuits in our platform. The source codes for Arduino and LabVIEW are available from the corresponding author upon request.

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

## Acknowledgements

This work was supported by IBS-R006-A1 and National Research Foundation of Korea (2017R1A2B3005842).

## Author contributions

K.W.C., S.J.K., J.K., S.Y.S., B.-S.K. and D.-H.K. conceived the idea and performed the experiments and analysis. L.W. and N.L. designed numerical simulation method on substrate characteristic. M.S. and T.H. synthesized and analyzed ceria-zirconia nano-particle. K.W.C., S.J.K., J.K., S.Y.S., W.H.L., L.W., M.S., N.L., T.H., B.-S.K. and D.-H.K. wrote the manuscript.

## Competing interests

The authors declare no competing interests.
