## [Peer Review File · Nature Communications]

Reviewers' comments:

Reviewer #1 (Remarks to the Author):

In this paper, Kyoung Won Cho et al. reported a highly integrated digital platform for cell cultures. They showed the device fabrication, individual device performance, scalability, wireless control, in vitro toxicity testing of drugs and cosmetics, and cell expansion. Overall, this is an impressive piece of work, and the authors have done very careful and thorough research! This research area, as far as this referee can tell, is a new direction from Kim lab, and the initial studies are convincing and show a lot of promise. This referee highly recommends its publication in Nature Communications. Only very minor comments are listed here for manuscript improvement:

(1) Consider to either expand some of the figure captions or main text. Many figure panels, such as Figure 3k, are only discussed very briefly. It would be great if slightly more details would be given.

(2) Many numbers, such as those in Figures 5a, 5b, 5j, 6c, can be deleted for clarity. It is hard to read them.

(3) This referee agrees that the technology presented here is transformative. However, can the authors point out the current limitations and approaches to address these limitations in the future.

(4) Scale bar in Figure 3k is missing.

Reviewer #2 (Remarks to the Author):

This manuscript presents a large-scale, integrated and smart cell-culture platform. Although a lot of work have been done to demonstrate the ability of this system used for mass cell culture, we think the work is lack of novelty. We have some reasons to reject this manuscript as shown in the following:

1. This manuscript introduces a large-scale, integrated and smart cell-culture platform. Actually, although the number of culture substrate layers in LISCCP can be increased up to 25 layers, all the sensors in one LISCCP monitor the same sample (the sensors are in the same culture chamber), we wonder if it is suitable for large-scale drug screening and toxicity testing. Maybe LISCCP can be expanded into a large number, we believe this method need a large number of cells (cell line is expensive), culture medium, and a lot of operation labors to culture cells.

2. As we see, there are four key parameters detected by your platform (LISCCP), cell impedance, pH, temperature and potassium ion (K⁺), but what we were provided were only the results of your experiments, not the principles of four types of sensors. So, we were wondering if it was the integration of the mature detection methods or did exist some innovations while you forget to clarify.

3. Here you mentioned "ultrathin sensors" in your abstract, but we did not see your relevant characterization or images presented to prove the thickness or other features.

4. The R² of your K⁺ fitting equation was missing in the Fig.2j.

5. In page 9, line 190-191, you mentioned 'these sensors can detect subtle pH changes between 6.95-7.95', does the detection range large enough in this case?

6. The frequency of your stimulation you used was 200 Hz, but the frequency-scanning result in Fig.2k showed no obvious difference between 200 Hz and others, so why chose this value as your final working frequency?

7. In Fig.3j, you compared the wireless measured pH and K⁺ with the results of the wired system, of which the detailed methods and specific instruments were not provided.

8. In fig.4k in page15 line301-305, 'the impedance decrease due to increased cell-to-cell electrical coupling via increased cell-cell contact at high cell concentration.' We think the impedance should be increased in this case.

9. In fig.4i-k, Can the authors describe and explain the results in more detail? The cell morphology can be described quantitatively? We are confused about the fluorescent image.

10. In fig. 4c, fig. 5a, the impedance of C2C12 reaches a maximum value at day 5 and day 7 respectively, why the results are different in the same experimental condition?

11. In figure 5g-I, the mechanism that electrical and thermal stimulations affect C2C12 proliferation and differentiation should be discussed or explained.
12. In fig. Sd8 and page 9 line 182-184, you mentioned 'the cell adhesion and subsequent proliferation increase further as the concentration of GO used for PLA coating increases', does it mean the higher concentration of GO, the better cell adhesion and proliferation? What is the best concentration of GO for cell culture?
13. In Fig. S18a you presented the graphs standing for the FEM analysis results of ROS, have you tried or compared those with the values monitored by other methods, like the gas chromatography or the Clark-type electrodes.
14. Here you marked "Digital Mass Culture" in your title with the control of layers from 1 to 25, so how did you manage to ensure the quality, numbers and distribution of cells in each layer after you seeded, and what about the passage as well as the harvesting step?
15. Following the question above, in what way did you measure the results of Live/Dead images because it seems hard to get the fluorescence images of the stacked chips in each layer respectively.
16. There were some fluctuations, which can be reduced by continuous medium circulation as you mentioned, in your pH and K⁺ monitoring curves (e.g. Fig. 4c), but the results seemed that it still existed in your Fig. 5d, so why it cannot be eliminated?
17. The platform you proposed seemed to still be in large size due to the peristaltic pump, and if we are not wrong, the replacement or supplement of medium needed for continuous circulation in that pump still requests manpower, and the industrial application of the toxicity testing or cosmetics production screening presented also require a pipette-procedures. So, it seems to be not that much labor-saving.

Reviewer #1:

Summary Comments: *In this paper, Kyoung Won Cho et al. reported a highly integrated digital platform for cell cultures. They showed the device fabrication, individual device performance, scalability, wireless control, in vitro toxicity testing of drugs and cosmetics, and cell expansion. Overall, this is an impressive piece of work, and the authors have done very careful and thorough research! This research area, as far as this referee can tell, is a new direction from Kim lab, and the initial studies are convincing and show a lot of promise. This referee highly recommends its publication in Nature Communications. Only very minor comments are listed here for manuscript improvement:*

Our response to summary comments: We appreciate the reviewer for constructive comments and valuable suggestions, which are helpful for improving our work. We have addressed the reviewer’s comments in a point-by-point manner and revised the manuscript accordingly. The revised parts are marked as red.

Comment #1: *Consider to either expand some of the figure captions or main text. Many figure panels, such as Figure 3k, are only discussed very briefly. It would be great if slightly more details would be given.*

Our response to comment #1: We thank the reviewer for the valuable comment. We have included more detailed explanation on the following figure panels in the revised manuscript.

Our modification to the manuscript:
(Figure 1i captions: in the revised manuscript)

Figure 1i | Photograph of a CO₂ incubator filled with multi-stacked LISCCPs (a total of 10) for digital mass cell culture.

(Figure 3k: in the revised manuscript)

Figure 3k | Infrared camera images of the heat generated and controlled by the thermal stimulators (TS) wirelessly. The number of TS to turn on can be controlled.

(Figure 4d-g: in the revised manuscript)

Figure 4d-g | Impedance mappings of hDFB (d), hMSC (e), C2C12 (f), and HL-1 (g). The blue color indicates the lowest impedance, whereas red color indicates the impedance in maximum.

(Figure 4h: in the revised manuscript)

Figure 4h | Growths of four types of cells as determined by the WST-8 assay. The absorbance is increased as the number of cells is increased.

Comment #2: Many numbers, such as those in Figures 5a, 5b, 5j, 6c, can be deleted for clarity. It is hard to read them.

Our response to comment #2: We thank the reviewer for the valuable comment. We have removed all the numbers that are not shown clearly in the impedance color maps.

Our modification to the manuscript:

(Figure 5a,b: in the revised manuscript)

(Figure 5j: in the revised manuscript)

(Figure 6c: in the revised manuscript)

Comment #3: This referee agrees that the technology presented here is transformative. However, can the authors point out the current limitations and approaches to address these limitations in the future.

Our response to comment #3: We appreciate the reviewer for the constructive comment. We modified the discussion part in the revised manuscript to address the comment.

Our modification to the manuscript:

(Line 12, page 24: in the revised manuscript)

Line 14, page 24 → Line 12, page 24

“**Since** the PLA substrate layers are 3-D printed and designed like a Lego® block, the assembly of each layer as well as the disassembly for the separate use of each layer is facile.”

(Line 20, page 24: in the revised manuscript)

“**Although the culture medium circulation system allows the cell culture of up to 25 layers, integration of automated medium perfusion facility in the culture system may increase the culture substrate number more than 25 layers. The integration of the culture medium perfusion facility can also prevent fluctuation of pH and ion concentrations within the culture medium. Meanwhile, the miniaturization of the wireless system in LISCCP can further increase the space efficiency so that more LISCCPs can be placed in an incubator. In the future study, the current Arduino-based wireless system can be replaced by the customized wireless chip, which will decrease the overall system size further.**”

Comment #4: Scale bar in Figure 3k is missing.

Our response to comment #4: We thank the reviewer for the detailed comment. The scale bar for Figure 3k has been modified.

Our modification to the manuscript:

(Figure 3k: in the revised manuscript)

Thank you very much again for your insightful comments. We feel that these comments have helped to improve the quality of the manuscript significantly.

Reviewer #2:

Summary Comments: *This manuscript presents a large-scale, integrated and smart cell-culture platform. Although a lot of work have been done to demonstrate the ability of this system used for mass cell culture, we think the work is lack of novelty. We have some reasons to reject this manuscript as shown in the following:*

Our response to summary comments: We sincerely appreciate the reviewer for the detailed and insightful comments on our work that have improved the overall quality of our manuscript. The novelty of this study is in that the large-scale integration of multiple ultrathin sensor/stimulator arrays in a wirelessly controlled manner enables efficient, effective, and real-time monitoring and control of large scale culture of the anchorage-dependent cells. This also enables the transformation of conventional large-scale 2D cell culturing methods to efficient digital mass cell cultures, which has not been reported before. We have also addressed the reviewer's comments in a point-by-point manner and revised the manuscript accordingly.

Comment #1: *This manuscript introduces a large-scale, integrated and smart cell-culture platform. Actually, although the number of culture substrate layers in LISCCP can be increased u to 25 layers, all the sensors in one LISCCP monitor the same sample (the sensors are in the same culture chamber), we wonder if it is suitable for large-scale drug screening and toxicity testing. Maybe LISCCP can be expanded into a large number, we believe this method need a large number of cells (cell line is expensive), culture medium, and a lot of operation labors to culture cells.*

Our response to comment #1: We thank the reviewer for the comment. Each substrate layer contains sensors and stimulators for cells cultured on that specific layer. Therefore, cells on each layer can be independently monitored and stimulated. After the cell culture, each substrate layer can be separated and utilized as an independent *in vitro* platform for large-scale drug screening and toxicity testing. The number of cells and the quantity of culture medium required for drug screening or toxicity testing at the same scale would not be different between conventional methods and our system. However, operation labor for large-scale cell culture can be fairly reduced by our platform, since each culture layer in our system does not need to be taken outside of the incubator for individual sensing and stimulation as well as medium change. Also, observation for cells *via* optical microscopy after taking the culture system out of the incubator is not required. We have modified the text to clarify the advantages of our system in the revised manuscript.

Our modification to the manuscript:

(Line 9, page 24: in the revised manuscript)

“The integrated sensor arrays **on each substrate layer, which is stacked into multiple layers in a chamber of LISCCP**, allow thorough 3D monitoring of the electrophysiological and environmental conditions of cultured cells.”

(Line 16, page 25: in the revised manuscript)

“**Also, LISCCP does not need to be taken outside of the incubator for monitoring/stimulation of the cells or for change of culture medium.**”

Comment #2: As we see, there are four key parameters detected by your platform (LISCCP), cell impedance, pH, temperature and potassium ion (K^+), but what we were provided were only the results of your experiments, not the principles of four types of sensors. So, we were wondering if it was the integration of the mature detection methods or did exist some innovations while you forget to clarify.

Our response to comment #2: We thank the reviewer for the comment. Each sensor and stimulator has been developed previously. However, the idea for the large-scale integration of these sensors and stimulators in the ultrathin device arrays along with wireless units toward the digital mass cell culture has not been reported yet. Our LISCCP contains arrays of multiple sensors and stimulators so that the cultured cells can be controlled wirelessly and the massive monitoring data can be obtained easily. Therefore, we believe that such large-scale integration of ultrathin sensors and actuators with the wireless system suggests a novel direction for mass culture of anchorage-dependent cells.

Comment #3: Here you mentioned “ultrathin sensors” in your abstract, but we did not see your relevant characterization or images presented to prove the thickness or other features.

Our response to comment #3: We thank the reviewer for the constructive comment. The thickness of each layer including encapsulation layers are described in Method. The total thickness of our integrated sensors and stimulators including all the encapsulation layers is approximately $\sim 11 \mu\text{m}$. We have included a separate figure (Supplementary Figure S4) in Supplementary Information to show the scanning electron microscopic images of the cross-section and corresponding schematic illustrations for the thickness information of each device.

Our modification to the manuscript:
(Supplementary Figure S4: in the revised manuscript)

Supplementary Figure S4 | Thickness information of the integrated ultrathin sensors and stimulators. **a**, exploded view of sensors and stimulators with thickness information for each layer. The total thickness of the sensors and stimulators including the encapsulation layers is ~11 μm . **b-e**, scanning electron microscopic images of the cross-sections for (b) the device, (c) impedance sensor/electrical stimulator, (d) temperature sensor and thermal stimulator, and (e) pH sensor/ K^+ sensor.

(Line 2, page 6: in the revised manuscript)

“The total thickness of the device is ~ 11 μm . The thickness of each layer and the scanning electron microscopy (SEM) image are shown in Supplementary Fig. S4.”

Comment #4: The R^2 of your K^+ fitting equation was missing in the Fig.2j.

Our response to comment #4: We thank the reviewer for the detailed comment. We included the R^2 information for the potassium ion measurement in Figure 2j.

Our modification to the manuscript:

(Figure 2j: in the revised manuscript)

Comment #5: In page 9, line 190-191, you mentioned ‘these sensors can detect subtle pH changes between 6.95-7.95’, does the detection range large enough in this case?.

Our response to comment #5: We thank the reviewer for the comment. Our pH sensor can detect pH in the range of 4 to 10, which is shown in Fig. 2i left. In Fig 2i right, we measured pH between 6.95 and 7.95 to show that the sensor can also detect small changes in pH. We specifically chose that range because the pH of the culture medium is 7.4.

Comment #6: The frequency of your stimulation you used was 200 Hz, but the frequency-scanning result in Fig.2k showed no obvious difference between 200 Hz and others, so why chose this value as your final working frequency ?

Our response to comment #6: We thank the reviewer for the detailed comment. In the impedance measurement of cells, the domain of alternate current (AC) frequency determines how the current flows from one electrode to another. At the low frequency ($10^2 - 10^4$ Hz), most of the current flow through the intercellular space (*Biosensors* 8, (2018), reference 20 in the revised manuscript). In

other words, the resistance at the low frequency can be easily altered for proliferative cells covering the electrodes as well as for changes in cell morphology such as cell fusion and connexin formation. Therefore, we set the frequency of AC in the low frequency range of 200 Hz to record changes in cell morphological development. We have added sentences in the revised manuscript to explain this point.

Our modification to the manuscript:

(Line 10, page 12: in the revised manuscript)

“The alternate current at low frequency (e.g., range of 10^2 - 10^4 Hz) flows through the paracellular space of the cells²⁰. To obtain sensitive detection on changes in the cell morphology such as differentiation of C2C12, we set the frequency as 200 Hz for all wireless impedance measurements.”

(Line 38, page 33: in the revised manuscript)

“20. Robilliard, L. D. *et al.* The importance of multifrequency impedance sensing of endothelial barrier formation using ECIS technology for the generation of a strong and durable paracellular barrier. *Biosensors* **8**, (2018).”

Comment #7: In Fig.3j, you compared the wireless measured pH and K⁺ with the results of the wired system, of which the detailed methods and specific instruments were not provided.

Our response to comment #7: We thank the reviewer for the comment. As described in Method, we used the open circuit potential mode of the electrochemical analyzer (CH Instrument, Inc) to measure pH and K⁺ concentrations for the wired system. We have modified the text to clarify the measurement method in the revised manuscript.

Our modification to the manuscript:

(Line 17, page 12: in the revised manuscript)

“The wirelessly measured pH and K⁺ changes are compatible with measurement by a wired system in the open circuit potential mode using a commercial electrochemical analyzer (CH Instruments, Inc) (Fig. 3j).”

Comment #8: In fig.4k in page15 line301-305, 'the impedance decrease due to increased cell-to-cell electrical coupling via increased cell-cell contact at high cell concentration.' We think the impedance should be increased in this case.

Our response to comment #8: We thank the reviewer for the comment. Because cell bodies act as an insulator in comparison with the conductive culture medium, the impedance is increased as the cells proliferate and cover the electrodes on the culture substrate. In case of cardiomyocytes, however, the electrical cell-to-cell conductance is correlated to connexin 43 that presents in gap junction between cells in contact (*Am. J. Physiol. Circ. Physiol.* **302**, H443–H450 (2011), reference 27 in the revised manuscript). After the cells reach their confluency where the impedance is the maximum, the cell-cell electric coupling is formed *via* connexin 43 and thus the impedance is slightly decreased. We modified the text to explain this point more clearly.

Our modification to the manuscript:

(Line 13, page 16: in the revised manuscript)

“More connections within adjacent cells increases the electrical pathway, which slightly decreases intercellular impedance.”

Comment #9: In fig.4i-k, Can the authors describe and explain the results in more detail? The cell morphology can be described quantificationally? We are confused about the fluorescent image.

Our response to comment #9: We thank the reviewer for the comment. F-actin staining of C2C12 cells in Fig. 4i (top) shows the proliferation of C2C12. Myosin heavy chain staining of C2C12 cells in Fig. 4i (bottom) shows that the C2C12 cells form into multi-nucleated and elongated cells during the differentiation (*ACS Nano* **9**, 2677-2688 (2015), reference 4 in the revised manuscript). Figure 4k (top) shows F-actin staining images of HL-1 cells in which F-actin is increased as the cells proliferate. Connexin 43 (Cx 43, green) is stained in Fig. 4k (bottom) to show the increased level of Cx 43 during the differentiation of HL-1 cells (*Am. J. Physiol. Circ. Physiol.* **302**, H443–H450 (2011), reference 27 in the revised manuscript). We have also included another figure (Supplementary Figure S17) in Supplementary Information to quantitatively describe the myotube and Cx 43 areas and the related information is described in Method.

Our modification to the manuscript:

(Supplementary Figure S17: in the revised manuscript)

Supplementary Figure S17 | Quantitative descriptions on immunostaining images (Figure 4i,k) of C2C12 and HL-1 differentiations. a-b, (a) F-actin- and (b) myosin heavy chain-positive areas in the C2C12 culture. **c-d,** (c) F-actin- and (d) connexin 43-positive area of the HL-1 culture. (n=4, mean±s.d., *P<0.05, **P<0.01, ***P<0.001, ns, not significant, ANOVA with Bonferroni’s post-test)

(Line 2, page 15: in the revised manuscript)

“i, Fluorescent microscopy images of F-actin-stained C2C12 cells during proliferation (top) and myosin heavy chain staining of C2C12 cells showing myotube formation of the cells during differentiation (bottom).”

(Line 5, page 16: in the revised manuscript)

“Immunostaining of C2C12 shows that the F-actin of cell bodies covers the electrodes as the cells proliferate initially, while myotubes and multi-nucleus cells are observed during differentiation (Fig. 4i). The muscle-specific gene markers such as myogenin and MyoD are also upregulated after the induction of differentiation (Fig. 4j).”

(Line 12, page 16: in the revised manuscript)

“The increased expression of connexin 43, a gap junction protein of cardiomyocytes in the differentiation stage²⁷, is shown in Fig. 4k.”

(Line 15, page 16: in the revised manuscript)

“The increased expression of myotube for C2C12 and connexin 43 for HL-1 are quantitatively described in Supplementary Fig. S17.”

(Line 20, page 31: in the revised manuscript)

“For the quantification of the F-actin-, myosin heavy chain- and Cx43-positive area, the positive area was counted via the percentage relative to the total image area using ImageJ software (n=4).”

Comment #10: In fig. 4c, fig. 5a, the impedance of C2C12 reaches a maximum value at day 5 and day 7 respectively, why the results are different in the same experimental condition?

Our response to comment #10: We appreciate the reviewer for the detailed comment. After proliferation, C2C12 mouse skeletal muscle cells can form elongated and multi-nucleated cells even in the growth medium containing fetal bovine serum. But the differentiation culture medium containing horse serum can promote the differentiation of C2C12 further. In the experiment of Fig. 4c, we used the differentiation medium to promote more myotube formation. On the other hand, we did not use the differentiation medium on 5-layer cell culture experiments in Fig. 5 because we intended to remove the effect of the differentiation medium in order to see only electrical/thermal stimulation effect on the enhanced proliferation and differentiation of C2C12. We have now corrected the description in the text and clarified the experimental condition in Method.

Our modification to the manuscript:

(Line 1, page 30: in the revised manuscript)

“To monitor both proliferation and differentiation of C2C12 on the single-layer platform, we cultured the cells with differentiation medium. For C2C12 culture on 5-layered LISCCPs, no differentiation medium was treated to see only electrical/thermal stimulation effect on the enhanced proliferation and differentiation of C2C12 cells.”

Comment #11: In figure 5g-I, the mechanism that electrical and thermal stimulations affect C2C12 proliferation and differentiation should be discussed or explained.

Our response to comment #11: We thank the reviewer for the detailed comment. The effects of electrical and thermal stimulation on various types of cells have been widely studied. Many studies have reported enhanced proliferation and differentiation of C2C12 via electrical/thermal stimulations. Electrical stimulation applied onto the cells changes the resting transmembrane

potential of the cells, which causes the upregulated expression of growth factors and myokines (*J. Cell. Physiol.* **233**, 1860-1876 (2018), reference 31 in the revised manuscript). Also, mild heat stimulation to C2C12 induces mitochondrial biogenesis and upregulates AMP-activated protein kinase activity (*J. Appl. Physiol.* **112**, 354-361 (2012), reference 32 in the revised manuscript). In this study, we used the C2C12 cell line as an exemplary case to show that the electrical/thermal stimulators in our LISCCP can be utilized for such stimulation studies.

Our modification to the manuscript:

(Line 20, page 19: in the revised manuscript)

“LISCCP can promote cellular proliferation and differentiation *via* electrical/thermal stimulations in a wireless manner. Electrical stimulation alters the resting transmembrane potential, which upregulates the expression of growth factors³¹. Thermal stimulation induces mitochondrial biogenesis and enhance AMP-activated protein kinase activity³².”

(Line 1, page 35: in the revised manuscript)

“31. Love, M. R., Palee, S., Chattipakorn, S. C. & Chattipakorn, N. Effects of electrical stimulation on cell proliferation and apoptosis. *J. Cell. Physiol.* **233**, 1860–1876 (2018).”

(Line 3, page 35: in the revised manuscript)

“32. Liu, C.-T. & Brooks, G. A. Mild heat stress induces mitochondrial biogenesis in C2C12 myotubes. *J. Appl. Physiol.* **112**, 354–361 (2012).”

Comment #12: In fig.Sd8 and page 9 line 182-184, you mentioned ‘the cell adhesion and subsequent proliferation increase further as the concentration of GO used for PLA coating increases’, does it mean the higher concentration of GO, the better cell adhesion and proliferation? What is the best concentration of GO for cell culture?

Our response to comment #12: We appreciate the reviewer for the comment. It has been known that carbon-based nanomaterials, especially graphene oxide (GO), provide physiochemical properties that are comparable to those of natural extracellular matrix (*J. Biomed. Mater. Res. - Part A* **101**, 3520–3530 (2013), reference 16 in the revised manuscript). The enhanced cellular adhesion can be observed on spray-coated graphene oxide nanosheets, as compared to the bare glass or polystyrene substrates. We conducted additional studies (Supplementary Figure S9e) to ascertain the optimum concentration of GO for the cell culture and modified the Figure in Supplementary Information to show the optimum GO concentration.

Our modification to the manuscript:

(Supplementary Figure S9e: in the revised manuscript)

Supplementary Figure S9e | the number of C2C12 cells after 3-day culture on PLA coated with GO of a wide range of concentrations to examine the effect of GO concentration on the cell proliferation. (n=8, Whiskers: min to max, *P<0.05, **P<0.01, ***P<0.001 versus zero concentration, ANOVA with Bonferroni's post-test)

(Line 17, page 9: in the revised manuscript)

“The cell proliferation increases as the GO concentration increases, but saturates after the GO concentration reaches 200 mg/100mL ethanol (Supplementary Fig. S9e).”

Comment #13: In Fig.S18a you presented the graphs standing for the FEM analysis results of ROS, have you tried or compared those with the values monitored by other methods, like the gas chromatography or the Clark-type electrodes.

Our response to comment #13: We thank the reviewer for the comment. The finite-element method analysis in Supplementary Fig. S18 shows the distribution of dissolved oxygen in the culture medium. The concentration of dissolved oxygen decreases as the depth of culture medium increases, which has been known in references (*e.g.*, *Hypoxia* **3**, 35 (2015), reference 1 in the Supplementary text). We have not experimentally measured the dissolved oxygen concentration in the culture medium. We conducted the simulation study on the dissolved oxygen to support that the culture medium circulation enhanced the cellular viability on 5-layer and even 25-layered substrate layers.

Our modification to the manuscript:

(Line 5, page 2: Supplementary text in the revised manuscript)

“Cells adherent on culture substrates in cell culture experience a lower concentration of dissolved oxygen as the depth of the culture medium increases¹.”

(Line 5, page 4: Supplementary text in the revised manuscript)

1. Wenger, R. H., Kurtcuoglu, V., Scholz, C. C., Marti, H. H., & Hoogewijs, D. Frequently asked questions in hypoxia research. *Hypoxia* **3**, 35 (2015)

Comment #14: Here you marked “Digital Mass Culture” in your title with the control of layers from 1 to 25, so how did you manage to ensure the quality, numbers and distribution of cells in each layer after you seeded, and what about the passage as well as the harvesting step?

Our response to comment #14: We thank the reviewer for the comment. Each substrate layer contains 36 perforations whose area is 1 mm². The height of the space between two substrate layers is 1 mm. Therefore, cells suspended in the culture medium can be distributed homogeneously in stacked substrate layers in the cell seeding process. The cell counting results in Fig. 5I shows that the cell number is similar between the top (Layer 1) and the bottom (Layer 5) layer on day 1. We can monitor the cell condition using the sensor array. Each substrate layer can be disassembled, and cells can be harvested independently. When entire cells in all layers are needed to be harvested at the same time, trypsin can be treated through the inlet tube.

Our modification to the manuscript:

(Line 7, page 6: in the revised manuscript)

“The 3D-printed¹² PLA substrate is designed to have multiple perforations (1 × 1 mm) for easy penetration of the culture medium and for distribution of seeded cells throughout multiple layers, protruding hemispheres (D = 1 mm) to increase the surface area, and stackable pillars and holes at edges for multilayer assembly (Supplementary Fig. S6).”

Comment #15: Following the question above, in what way did you measure the results of Live/Dead images because it seems hard to get the fluorescence images of the stacked chips in each layer respectively.

Our response to comment #15: We thank the reviewer for the comment. The Live Dead assay on each substrate layer was conducted as a control experiment to compare the cell viability obtained by the impedance monitoring data with that obtained by the Live Dead assay. Thus, in the real cell culture using LISCCPs, the Live Dead assay will not be conducted. Since the multiple substrate layers are stacked together like Lego blocks, the multiple substrate layers can be simply disassembled into a single layer to conduct the Live Dead assay. We have now modified the Method.

Our modification to the manuscript:

(Line 18, page 30: in the revised manuscript)

“The multiple substrate layers were disassembled into a single layers to conduct biological assays such as cell counting, cell viability determination, and the Live and Dead assay.”

Comment #16: There were some fluctuations, which can be reduced by continuous medium circulation as you mentioned, in your pH and K⁺ monitoring curves (e.g. Fig.4c), but the results seemed that it still existed in your Fig.5d, so why it cannot be eliminated?

Our response to comment #16: We thank the reviewer for the valuable comment. The pH decreases during cell culture due to lactic acid production by cell metabolism. As cells proliferate, the decrease in pH is accelerated. In Fig. 5d, pH decreased as the cultured cells produced lactic acid over culture time. In the same figure, pH increased when fresh culture medium is replaced every other day. Thus, the fluctuation of pH is inevitable in Fig. 5d. The culture medium circulation improves the mass transfer within the culture medium, which helps to reduce the extent of pH fluctuation. But it cannot eliminate the fluctuation since the medium is changed periodically. In the future study, a continuous medium perfusion facility may eliminate pH fluctuation completely. We have added this perspective for the future study in the revised manuscript.

Our modification to the manuscript:

(Line 20, page 24: in the revised manuscript)

“Although the culture medium circulation system allows the cell culture of up to 25 layers, integration of automated medium perfusion facility in the culture system may increase the culture substrate number more than 25 layers. The integration of the culture medium perfusion facility can also prevent fluctuation of pH and ion concentrations within the culture medium.”

Comment #17: The platform you proposed seemed to still be in large size due to the peristaltic pump, and if we are not wrong, the replacement or supplement of medium needed for continuous circulation in that pump still requests manpower, and the industrial application of the toxicity testing or cosmetics production screening presented also require a pipette-procedures. So, it seems to be not that much labor-saving.

Our response to comment #17: We appreciate the reviewer for the comment. As the reviewer mentioned, the size of the peristaltic pump itself is large. Yet, the pump can reside outside of an incubator, while the pump is connected to LISCCP *via* tubes. In this study, we used a single-channel peristaltic pump, but there are multi-channel pumps that can connect multiple lines of tubes. The issue of culture medium replacement can be solved easily, especially in industry, by the installation of tanks for fresh medium feed and used medium harvest. We are looking forward to implementing the automated perfusion facility in future study.

In Figure 6, the pipette procedures were used to demonstrate treatment of drug candidates. We intended to demonstrate the real time monitoring of cells after treatment with drug candidates, as a demonstration for the *in vitro* toxicity testing platform. In many industrial laboratories, there already are robotic arms that can load drug candidates automatically. Alternatively, a large number of drug candidates can be applied to each platform through silicon tubes in the multi-channel pumping facilities. Therefore, the current LISCCP provides the key breakthroughs toward labor-saving in the large-scale cell culture.

Our modification to the manuscript:

(Line 24, page 24: in the revised manuscript)

“Meanwhile, the miniaturization of the wireless system in LISCCP can further increase the space efficiency so that more LISCCPs can be placed in an incubator. In the future study, the current Arduino-based wireless system can be replaced by the customized wireless chip, which will decrease the overall system size further.”

(Line 17, page 24: in the revised manuscript)

“The peristaltic pump of this system can reside outside of an incubator, while the pump is connected to LISCCP *via* the inlet and outlet tubes.”

(Line 10, page 25: in the revised manuscript)

“Moreover, a large number of drug candidates can be automatically applied to each platform of LISCCP through silicon tubes, if the multi-channel pumping facilities are installed.”

Thank you very much again for your insightful comments. We feel that these comments have helped to improve the quality of the manuscript significantly.

REVIEWERS' COMMENTS:

Reviewer #1 (Remarks to the Author):

The authors have done a very nice job in the revision. Publication is now recommended.

Reviewer #2 left no comments to the author.

Our response to the reviewers' comments

We sincerely appreciate the reviewers for their time and many valuable comments, which have improved the overall quality of our manuscript.

Reviewer #1: *The authors have done a very nice job in the revision. Publication is now recommended.*

Our Response: We thank the reviewer for the insightful suggestions and other detailed comments.

Reviewer #2: *left no comments to the author.*

Our Response: We appreciate the reviewer for reviewing our work. Many valuable comments from the reviewer have helped us to improve the quality of our work significantly.